# On Markov Chain Gradient Descent[*]

**Tao Sun**
College of Computer
National University of Defense Technology
Changsha, Hunan 410073, China
nudtsuntao@163.com

**Yuejiao Sun**
Department of Mathematics
University of California, Los Angeles
Los Angeles, CA 90095, USA
sunyj@math.ucla.edu

**Wotao Yin**
Department of Mathematics
University of California, Los Angeles
Los Angeles, CA 90095, USA
wotaoyin@math.ucla.edu

## Abstract

Stochastic gradient methods are the workhorse (algorithms) of large-scale optimization problems in machine learning, signal processing, and other computational sciences and engineering. This paper studies Markov chain gradient descent, a variant of stochastic gradient descent where the random samples are taken on the trajectory of a Markov chain. Existing results of this method assume convex objectives and a reversible Markov chain and thus have their limitations. We establish new non-ergodic convergence under wider step sizes, for nonconvex problems, and for non-reversible finite-state Markov chains. Nonconvexity makes our method applicable to broader problem classes. Non-reversible finite-state Markov chains, on the other hand, can mix substatially faster. To obtain these results, we introduce a new technique that varies the mixing levels of the Markov chains. The reported numerical results validate our contributions.

## 1   Introduction

In this paper, we consider a stochastic minimization problem. Let $\Xi$ be a statistical sample space with probability distribution $\Pi$ (we omit the underlying $\sigma$-algebra). Let $X \subseteq \mathbb{R}^n$ be a closed convex set, which represents the parameter space. $F(\cdot; \xi) : X \to \mathbb{R}$ is a closed convex function associated with $\xi \in \Xi$. We aim to solve the following problem:

$$\underset{x \in X \subseteq \mathbb{R}^n}{\text{minimize}} \;\; \mathbb{E}_\xi\big(F(x; \xi)\big) = \int_\Pi F(x, \xi) d\Pi(\xi). \tag{1}$$

A common method to minimize (1) is Stochastic Gradient Descent (SGD) [11]:

$$x^{k+1} = \mathbf{Proj}_X\big(x^k - \gamma_k \partial F(x^k; \xi^k)\big), \quad \text{samples } \xi^k \overset{\text{i.i.d}}{\sim} \Pi. \tag{2}$$

However, for some problems and distributions, direct sampling from $\Pi$ is expensive or impossible, and it is possible that the sample space $\Xi$ is not explicitly known. In these cases, it can be much cheaper to sample by following a Markov chain that has a desired equilibrium distribution $\Pi$.

---

[*]The work is supported in part by the National Key R&D Program of China 2017YFB0202902, China Scholarship Council. The work of Y. Sun and W. Yin was supported in part by AFOSR MURI FA9550-18-1-0502, NSF DMS-1720237, NSFC 11728105, and ONR N000141712162.

To be concrete, imagine solving problem (1) with a discrete space $\Xi := \{x \in \{0,1\}^n \mid \langle a, x \rangle \leq b\}$, where $a \in \mathbb{R}^n$ and $b \in \mathbb{R}$, and the uniform distribution $\Pi$ over $\Xi$. A straightforward way to obtain a uniform sample is iteratively randomly sampling $x \in \{0,1\}^n$ until the constraint $\langle a, x \rangle \leq b$ is satisfied. Even if the feasible set is small, it may take up to $O(2^n)$ iterations to get a feasible sample. Instead, one can sample a trajectory of a Markov chain described in [4]; to obtain a sample $\varepsilon$-close to the distribution $\Pi$, one only needs $\log(\frac{\sqrt{|\Xi|}}{\varepsilon}) \exp(O(\sqrt{n}(\log n)^{\frac{5}{2}}))$ samples [2], where $|\Xi|$ is the cardinality of $\Xi$. This presents a signifant saving in sampling cost.

Markov chains also naturally arise in some applications. Common examples are systems that evolve according to Markov chains, for example, linear dynamic systems with random transitions or errors. Another example is a distributed system in which every node locally stores a subset of training samples; to train a model using these samples, we can let a token that holds all the model parameters traverse the nodes following a random walk, so the samples are accessed according to a Markov chain.

Suppose that the Markov chain has a stationary distribution $\Pi$ and a finite mixing time $T$, which is how long a random trajectory needs to be until its current state has a distribution that roughly matches $\Pi$. A larger $T$ means a closer match. Then, in order to run one iteration of (2), we can generate a trajectory of samples $\xi^1, \xi^2, \xi^3, \ldots, \xi^T$ and only take the last sample $\xi := \xi^T$. To run another iteration of (2), we repeat this process, i.e., sample a new trajectory $\xi^1, \xi^2, \xi^3, \ldots, \xi^T$ and take $\xi := \xi^T$.

Clearly, sampling a long trajectory just to use the last sample wastes a lot of samples, especially when $T$ is large. But, this may seem necessary because $\xi^t$, for all small $t$, have large biases. After all, it can take a long time for a random trajectory to explore all of the space, and it will often double back and visit states that it previously visited. Furthermore, it is also difficult to choose an appropriate $T$. A small $T$ will cause large bias in $\xi^T$, which slows the SGD convergence and reduces its final accuracy. A large $T$, on the other hand, is wasteful especially when $x^k$ is still far from convergence and some bias does not prevent (2) to make good progress. Therefore, $T$ should increase adaptively as $k$ increases — this makes the choice of $T$ even more difficult.

So, why waste samples, why worry about $T$, and why not just apply every sample immediately in stochastic gradient descent? This approach has appeared in [5, 6], which we call the Markov Chain Gradient Descent (MCGD) algorithm for problem (1):

$$x^{k+1} = \mathbf{Proj}_X\big(x^k - \gamma_k \hat{\nabla} F(x^k; \xi^k)\big), \tag{3}$$

where $\xi^0, \xi^1, \ldots$ are samples on a Markov chain trajectory and $\hat{\nabla} F(x^k; \xi^k) \in \partial F(x^k; \xi^k)$ is a subgradient.

Let us examine some special cases. Suppose the distribution $\Pi$ is supported on a set of $M$ points, $y^1, \ldots, y^M$. Then, by letting $f_i(x) := M \cdot \mathrm{Prob}(\xi = y^i) \cdot F(x, y^i)$, problem (1) reduces to the finite-sum problem:

$$\underset{x \in X \subseteq \mathbb{R}^d}{\mathrm{minimize}} \, f(x) \equiv \frac{1}{M} \sum_{i=1}^{M} f_i(x). \tag{4}$$

By the definition of $f_i$, each state $i$ has the uniform probability $1/M$. At each iteration $k$ of MCGD, we have

$$x^{k+1} = \mathbf{Proj}_X\big(x^k - \gamma_k \hat{\nabla} f_{j_k}(x^k)\big), \tag{5}$$

where $(j_k)_{k \geq 0}$ is a trajectory of a Markov chain on $\{1, 2, \ldots, M\}$ that has a uniform stationary distribution. Here, $(\xi^k)_{k \geq 0} \subseteq \Pi$ and $(j_k)_{k \geq 0} \subseteq [M]$ are two different, but related Markov chains. Starting from a deterministic and arbitrary initialization $x^0$, the iteration is illustrated by the following diagram:

$$
\begin{array}{ccccccc}
j_0 & \longrightarrow & j_1 & \longrightarrow & j_2 & \longrightarrow & \ldots \\
\downarrow & & \downarrow & & \downarrow & & \\
x^0 \longrightarrow & x^1 & \longrightarrow & x^2 & \longrightarrow & x^3 & \longrightarrow \ldots
\end{array} \tag{6}
$$

In the diagram, given each $j_k$, the next state $j_{k+1}$ is statistically independent of $j_{k-1}, \ldots, j_0$; given $j_k$ and $x^k$, the next iterate $x^{k+1}$ is statistically independent of $j_{k-1}, \ldots, j_0$ and $x^{k-1}, \ldots, x^0$.

Another application of MCGD involves a network: consider a strongly connected graph $\mathcal{G} = (\mathcal{V}, \mathcal{E})$ with the set of vertices $\mathcal{V} = \{1, 2, \ldots, M\}$ and set of edges $\mathcal{E} \subseteq \mathcal{V} \times \mathcal{V}$. Each node $j \in \{1, 2, \ldots, M\}$ possess some data and can compute $\nabla f_j(\cdot)$. To run MCGD, we employ a token that carries the variable $x$, walking randomly over the network. When it reaches a node $j$, node $j$ reads $x$ form the token and computes $\nabla f_j(\cdot)$ to update $x$ according to (5). Then, the token walks away to a random neighbor of node $j$.

## 1.1 Numerical tests

We present two kinds of numerical results. The first one is to show that MCGD uses fewer samples to train both a convex model and a nonconvex model. The second one demonstrates the advantage of the faster mixing of a non-reversible Markov chain. Our results on nonconvex objective and non-reversible chains are new.

*1. Comparision with SGD*
Let us compare:

1. MCGD (3), where $j_k$ is taken from one trajectory of the Markov chain;
2. SGD$T$, for $T = 1, 2, 4, 8, 16, 32$, where each $j_k$ is the $T$th sample of a fresh, independent trajectory. All trajectories are generated by starting from the same state 0.

To compute $T$ gradients, SGD$T$ uses $T$ times as many samples as MCGD. We did not try to adapt $T$ as $k$ increases because there lacks a theoretical guidance.

In the first test, we recover a vector $u$ from an auto regressive process, which closely resembles the first experiment in [1]. Set matrix A as a subdiagonal matrix with random entries $A_{i,i-1} \overset{\text{i.i.d}}{\sim} \mathcal{U}[0.8, 0.99]$. Randomly sample a vector $u \in \mathbb{R}^d$, $d = 50$, with the unit 2-norm. Our data $(\xi_t^1, \xi_t^2)_{t=1}^\infty$ are generated according to the following auto regressive process:

$$\xi_t^1 = A\xi_{t-1}^1 + e_1 W_t, \ W_t \overset{\text{i.i.d}}{\sim} N(0, 1)$$

$$\bar{\xi}_t^2 = \begin{cases} 1, & \text{if } \langle u, \xi_t^1 \rangle > 0, \\ 0, & \text{otherwise}; \end{cases}$$

$$\xi_t^2 = \begin{cases} \bar{\xi}_t^2, & \text{with probability } 0.8, \\ 1 - \bar{\xi}_t^2, & \text{with probability } 0.2. \end{cases}$$

Clearly, $(\xi_t^1, \xi_t^2)_{t=1}^\infty$ forms a Markov chain. Let $\Pi$ denote the stationary distribution of this Markov chain. We recover $u$ as the solution to the following problem:

$$\underset{x}{\text{minimize}} \ \ \mathbb{E}_{(\xi^1, \xi^2) \sim \Pi} \ell(x; \xi^1, \xi^2).$$

We consider both convex and nonconvex loss functions, which were not done before in the literature. The convex one is the logistic loss

$$\ell(x; \xi^1, \xi^2) = -\xi^2 \log(\sigma(\langle x, \xi^1 \rangle)) - (1 - \xi^2) \log(1 - \sigma(\langle x, \xi^1 \rangle)),$$

where $\sigma(t) = \frac{1}{1+\exp(-t)}$. And the nonconvex one is taken as

$$\ell(x; \xi^1, \xi^2) = \frac{1}{2}(\sigma(\langle x, \xi^1 \rangle) - \xi^2)^2$$

from [7]. We choose $\gamma_k = \frac{1}{k^q}$ as our stepsize, where $q = 0.501$. This choice is consistently with our theory below.

Our results in Figure 1 are surprisingly positive on MCGD, more so to our expectation. As we had expected, MCGD used significantly fewer total samples than SGD on every $T$. But, it is surprising that MCGD did not need even more gradient evaluations. Randomly generated data must have helped homogenize the samples over the different states, making it less important for a trajectory to converge. It is important to note that SGD1 and SGD2, as well as SGD4, in the nonconvex case, stagnate at noticeably lower accuracies because their $T$ values are too small for convergence.

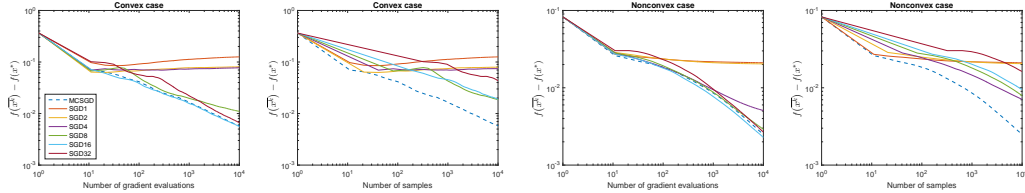

Figure 1: Comparisons of MCGD and SGD$T$ for $T = 1, 2, 4, 8, 16, 32$. $\overline{x^k}$ is the average of $x^1, \ldots, x^k$.

## 2. Comparison of reversible and non-reversible Markov chains

We also compare the convergence of MCGD when working with reversible and non-reversible Markov chains (the definition of reversibility is given in next section). As mentioned in [14], transforming a reversible Markov chain into non-reversible Markov chain can significantly accelerate the mixing process. This technique also helps to accelerate the convergence of MCGD.

In our experiment, we first construct an undirected connected graph with $n = 20$ nodes with edges randomly generated. Let $G$ denote the adjacency matrix of the graph, that is,

$$G_{i,j} = \begin{cases} 1, & \text{if } i, j \text{ are connected;} \\ 0, & \text{otherwise.} \end{cases}$$

Let $d_{\max}$ be the maximum number of outgoing edges of a node. Select $d = 10$ and compute $\beta^* \sim \mathcal{N}(0, I_d)$. The transition probability of the reversible Markov chain is then defined by, known as Metropolis-Hastings markov chain,

$$P_{i,j} = \begin{cases} \frac{1}{d_{\max}}, & \text{if } j \neq i, G_{i,j} = 1; \\ 1 - \frac{\sum_{j \neq i} G_{i,j}}{d_{\max}}, & \text{if } j = i; \\ 0, & \text{otherwise.} \end{cases}$$

Obviously, $P$ is symmetric and the stationary distribution is uniform. The non-reversible Markov chain is constructed by adding cycles. The edges of these cycles are directed and let $V$ denote the adjacency matrix of these cycles. If $V_{i,j} = 1$, then $V_{j,i} = 0$. Let $w_0 > 0$ be the weight of flows along these cycles. Then we construct the transition probability of the non-reversible Markov chain as follows,

$$Q_{i,j} = \frac{W_{i,j}}{\sum_l W_{i,l}},$$

where $W = d_{\max}P + w_0 V$. See [14] for an explanation why this change makes the chain mix faster.

In our experiment, we add 5 cycles of length 4, with edges existing in $G$. $w_0$ is set to be $\frac{d_{\max}}{2}$. We test MCGD on a least square problem. First, we select $\beta^* \sim \mathcal{N}(0, I_d)$; and then for each node $i$, we generate $x_i \sim \mathcal{N}(0, I_d)$, and $y_i = x_i^T \beta^*$. The objective function is defined as,

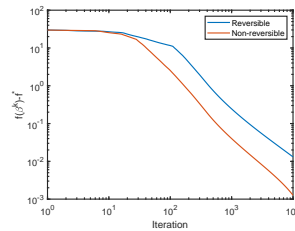

Figure 2: Comparison of reversible and irreversible Markov chains. The second largest eigenvalues of reversible and non-reversible Markov chains are 0.75 and 0.66 respectively.

$$f(\beta) = \frac{1}{2} \sum_{i=1}^{n} (x_i^T \beta - y_i)^2.$$

The convergence results are depicted in Figure 2.

### 1.2 Known approaches and results

It is more difficult to analyze MCGD due to its biased samples. To see this, let $p_{k,j}$ be the probability to select $\nabla f_j$ in the $k$th iteration. SGD's uniform probability selection ($p_{k,j} \equiv \frac{1}{M}$) yields an unbiased

gradient estimate

$$\mathbb{E}_{j_k}(\nabla f_{j_k}(x^k)) = C\nabla f(x^k) \tag{7}$$

for some $C > 0$. However, in MCGD, it is possible to have $p_{k,j} = 0$ for some $k, j$. Consider a "random walk". The probability $p_{j_k,j}$ is determined by the current state $j_k$, and we have $p_{j_k,i} > 0$ only for $i \in \mathcal{N}(j_k)$ and $p_{j_k,i} = 0$ for $i \notin \mathcal{N}(j_k)$, where $\mathcal{N}(j_k)$ denotes the neighborhood of $j_k$. Therefore, we no longer have (7).

All analyses of MCGD must deal with the biased expectation. Papers [6, 5] investigate the conditional expectation $\mathbb{E}_{j_{k+\tau}|j_k}(\nabla f_{j_{k+\tau}}(x^k))$. For a sufficiently large $\tau \in \mathbb{Z}^+$, it is sufficiently close to $\frac{1}{M}\nabla f(x^k)$ (but still different). In [6, 5], the authors proved that, to achieve an $\epsilon$ error, MCGD with stepsize $O(\epsilon)$ can return a solution in $O(\frac{1}{\epsilon^2})$ iteration. Their error bound is given in the ergodic sense and using $\liminf$. The authors of [10] proved a $\liminf f(x^k)$ and $\mathbb{E}\text{dist}^2(x^k, X^*)$ have almost sure convergence under diminishing stepsizes $\gamma_k = \frac{1}{k^q}, \frac{2}{3} < q \leq 1$. Although the authors did not compute any rates, we computed that their stepsizes will lead to a solution with $\epsilon$ error in $O(\frac{1}{\epsilon^{\frac{1}{1-q}}})$ iterations, for $\frac{2}{3} < q < 1$, and $O(e^{\frac{1}{\epsilon}})$ for $q = 1$. In [1], the authors improved the stepsizes to $\gamma_k = \frac{1}{\sqrt{k}}$ and showed ergodic convergence; in other words, to achieve $\epsilon$ error, it is enough to run MCGD for $O(\frac{1}{\epsilon^2})$ iterations. There is no non-ergodic result regarding the convergence of $f(x^k)$. It is worth mentioning that [10, 1] use time non-homogeneous Markov chains, where the transition probability can change over the iterations as long as there is still a finite mixing time. In [1], MCGD is generalized from gradient descent to mirror descent. In all these works, the Markov chain is required to be reversible, and all functions $f_i, i \in [M]$, are assumed to be convex. However, non-reversible chains can have substantially faster convergence and thus more numerically efficient.

## 1.3 Our approaches and results

In this paper, we improve the analyses of MCGD to non-reversible finite-state Markov chains and to nonconvex functions. The former allows us to have faster mixing, and the latter frequently appears in applications. Our convergence result is given in the non-ergodic sense though the rate results are still given the ergodic sense. It is important to mention that, in our analysis, the mixing time of the underlying Markov chain is not tied to a fixed mixing level but can vary to different levels. This is essential because MCGD needs time to reduce its objective error from its current value to a lower one, and this time becomes longer when the current value is lower since a more accurate Markov chain convergence and thus a longer mixing time are required. When $f_1, f_2, \ldots, f_M$ are all convex, we allow them to be non-differentiable and MCGD to use subgradients, provided that $X$ is bounded. When any of them is nonconvex, we assume $X$ is the full space and $f_1, f_2, \ldots, f_M$ are differentiable with bounded gradients. The bounded-gradient assumption is due to a technical difficulty associated with nonconvexity.

Specifically, in the convex setting, we prove $\lim_k \mathbb{E}f(x^k) = f^*$ (minimum of $f$ over $X$) for both exact and inexact MCGD with stepsizes $\gamma_k = \frac{1}{k^q}, \frac{1}{2} < q < 1$. The convergence rates of MCGD with exact and inexact subgradient computations are presented. The first analysis of nonconvex MCGD is also presented with its convergence given in the expectation of $\|\nabla f(x^k)\|$. These results hold for non-reversible finite-state Markov chains and can be extended to time non-homogeneous Markov chain under extra assumptions [10, Assumptions 4 and 5] and [1, Assumption C], which essentially ensure finite mixing.

Our results for finite-state Markov chains are first presented in Sections 3 and 4. They are extended to continuous-state reversible Markov chains in Section 5.

Some novel results are are developed based on new techniques and approaches developed in this paper. To get the stronger results in general cases, we used the varying mixing time rather than fixed ones.

We list the possible extensions of MCGD that are not discussed in this paper. The first one is the accelerated versions including the Nesterov's acceleration and variance reduction schemes. The second one is the design and optimization of Markov chains to improve the convergence of MCGD.

## 2 Preliminaries

### 2.1 Markov chain

We recall some definitions, properties, and existing results about the Markov chain. Although we use the finite-state time-homogeneous Markov chain, results can be extended to more general chains under similar extra assumptions in [10, Assumptions 4, 5] and [1, Assumption C].

**Definition 1 (finite-state time-homogeneous Markov chain)** *Let $P$ be an $M \times M$-matrix with real-valued elements. A stochastic process $X_1, X_2, ...$ in a finite state space $[M] := \{1, 2, \ldots, M\}$ is called a time-homogeneous Markov chain with transition matrix $P$ if, for $k \in \mathbb{N}$, $i, j \in [M]$, and $i_0, i_1, \ldots, i_{k-1} \in [M]$, we have*

$$\mathbb{P}(X_{k+1} = j \mid X_0 = i_0, X_1 = i_1, \ldots, X_k = i) = \mathbb{P}(X_{k+1} = j \mid X_k = i) = P_{i,j}. \tag{8}$$

Let the probability distribution of $X_k$ be denoted as the non-negative row vector $\pi^k = (\pi_1^k, \pi_2^k, \ldots, \pi_M^k)$, that is, $\mathbb{P}(X_k = j) = \pi_j^k$. $\pi$ satisfies $\sum_{i=1}^M \pi_i^k = 1$. When the Markov chain is time-homogeneous, we have $\pi^k = \pi^{k-1} P$ and

$$\pi^k = \pi^{k-1} P = \cdots = \pi^0 P^k, \tag{9}$$

for $k \in \mathbb{N}$, where $P^k$ denotes the $k$th power of $P$. A Markov chain is irreducible if, for any $i, j \in [M]$, there exists $k$ such that $(P^k)_{i,j} > 0$. State $i \in [M]$ is said to have a period $d$ if $P_{i,i}^k = 0$ whenever $k$ is *not* a multiple of $d$ and $d$ is the greatest integer with this property. If $d = 1$, then we say state $i$ is aperiodic. If every state is aperiodic, the Markov chain is said to be aperiodic.

Any time-homogeneous, irreducible, and aperiodic Markov chain has a stationary distribution $\pi^* = \lim_k \pi^k = [\pi_1^*, \pi_2^*, \ldots, \pi_M^*]$ with $\sum_{i=1}^M \pi_i^* = 1$ and $\min_i \{\pi_i^*\} > 0$, and $\pi^* = \pi^* P$. It also holds that

$$\lim_k P^k = [(\pi^*); (\pi^*); \ldots; (\pi^*)] =: \Pi^* \in \mathbb{R}^{M \times M}. \tag{10}$$

The largest eigenvalue of $P$ is 1, and the corresponding left eigenvector is $\pi^*$.

**Assumption 1** *The Markov chain $(X_k)_{k \geq 0}$ is time-homogeneous, irreducible, and aperiodic. It has a transition matrix $P$ and has stationary distribution $\pi^*$.*

### 2.2 Mixing time

Mixing time is how long a Markov chain evolves until its current state has a distribution very close to its stationary distribution. The literature has a thorough investigation of various kinds of mixing times, with the majority for reversible Markov chains (that is, $\pi_i P_{i,j} = \pi_j P_{j,i}$). Mixing times of non-reversible Markov chains are discussed in [3]. In this part, we consider a new type of mixing time of non-reversible Markov chain. The proofs are based on basic matrix analysis. Our mixing time gives us a direct relationship between $k$ and the deviation of the distribution of the current state from the stationary distribution.

To start a lemma, we review some basic notions in linear algebra. Let $\mathbb{C}$ be the $n$-dimensional complex field. The modulus of a complex number $a \in \mathbb{C}$ is given as $|a|$. For a vector $x \in \mathbb{C}^n$, the $\ell_\infty$ and $\ell_2$ norms are defined as $\|x\|_\infty := \max_i |x_i|$, $\|x\|_2 := \sqrt{\sum_{i=1}^n |x_i|^2}$. For a matrix $A = [a_{i,j}] \in \mathbb{C}^{m \times n}$, its $\infty$-induced and Frobenius norms are $\|A\|_\infty := \max_{i,j} |a_{i,j}|$, $\|A\|_F := \sqrt{\sum_{i,j=1}^n |a_{i,j}|^2}$, respectively.

We know $P^k \to \Pi^*$, as $k \to \infty$. The following lemma presents a deviation bound for finite $k$.

**Lemma 1** *Let Assumption 1 hold and let $\lambda_i(P) \in \mathbb{C}$ be the $i$th largest eigenvalue of $P$, and*

$$\lambda(P) := \frac{\max\{|\lambda_2(P)|, |\lambda_M(P)|\} + 1}{2} \in [0, 1).$$

*Then, we can bound the largest entry-wise absolute value of the deviation matrix $\delta^k := \Pi^* - P^k \in \mathbb{R}^{M \times M}$ as*

$$\|\delta^k\|_\infty \leq C_P \cdot \lambda^k(P) \tag{11}$$

*for $k \geq K_P$, where $C_P$ is a constant that also depends on the Jordan canonical form of $P$ and $K_P$ is a constant that depends on $\lambda(P)$ and $\lambda_2(P)$. Their formulas are given in (45) and (46) in the Supplementary Material.*

**Remark 1** *If $P$ is symmetric, then all $\lambda_i(P)$'s are all real and nonnegative, $K_P = 0$, and $C_P \leq M^{\frac{3}{2}}$. Furthermore, (42) can be improved by directly using $\lambda_2^k(P)$ for the right side as*

$$\|\delta^k\|_\infty \leq \|\delta^k\|_F \leq M^{\frac{3}{2}} \cdot \lambda_2^k(P), \ \ k \geq 0.$$

## 3 Convergence analysis for convex minimization

This part considers the convergence of MCGD in the convex cases, i.e., $f_1, f_2, \ldots, f_M$ and $X$ are all convex. We investigate the convergence of scheme (5). We prove non-ergodic convergence of the expected objective value sequence under diminishing non-summable stepsizes, where the stepsizes are required to be "almost" square summable. Therefore, the convergence requirements are almost equal to SGD. This section uses the following assumption.

**Assumption 2** *The set $X$ is assumed to be convex and compact.*

Now, we present the convergence results for MCGD in the convex (but not necessarily differentiable) case. Let $f^*$ be the minimum value of $f$ over $X$.

**Theorem 1** *Let Assumptions 1 and 2 hold and $(x^k)_{k \geq 0}$ be generated by scheme (5). Assume that $f_i$, $i \in [M]$, are convex functions, and the stepsizes satisfy*

$$\sum_k \gamma_k = +\infty, \quad \sum_k \ln k \cdot \gamma_k^2 < +\infty. \tag{12}$$

*Then, we have*

$$\lim_k \mathbb{E} f(x^k) = f^*. \tag{13}$$

*Define*

$$\psi(P) := \max\{1, \frac{1}{\ln(1/\lambda(P))}\}.$$

*We have:*

$$\mathbb{E}(f(\overline{x^k}) - f^*) = O\Big(\frac{\psi(P)}{\sum_{i=1}^k \gamma_i}\Big), \tag{14}$$

*where $\overline{x^k} := \frac{\sum_{i=1}^k \gamma_i x^i}{\sum_{i=1}^k \gamma_i}$. Therefore, if we select the stepsize $\gamma_k = O(\frac{1}{k^q})$ as $\frac{1}{2} < q < 1$, we get the rate $\mathbb{E}(f(\overline{x^k}) - f^*) = O(\frac{\psi(P)}{k^{1-q}})$.*

*Furthermore, consider the inexact version of MCGD:*

$$x^{k+1} = \mathbf{Proj}_X\big(x^k - \gamma_k(\hat{\nabla} f_{j_k}(x^k) + e^k)\big), \tag{15}$$

*where the noise sequence $(e^k)_{k \geq 0}$ is arbitrary but obeys*

$$\sum_{k=2}^{+\infty} \frac{\|e^k\|^2}{\ln k} < +\infty. \tag{16}$$

*Then, for iteration (15), results (13) and (14) still hold; furthermore, if $\|e^k\| = O(\frac{1}{k^p})$ with $p > \frac{1}{2}$ and $\gamma_k = O(\frac{1}{k^q})$ as $\frac{1}{2} < q < 1$, the rate $\mathbb{E}(f(\overline{x^k}) - f^*) = O(\frac{\psi(P)}{k^{1-q}})$ also holds.*

The stepsizes requirement (12) is nearly identical to the one of SGD and subgradient algorithms. In the theorem above, we use the stepsize setting $\gamma_k = O(\frac{1}{k^q})$ as $\frac{1}{2} < q < 1$. This kind of stepsize requirements also works for SGD and subgradient algorithms. The convergence rate of MCGD is $O(\frac{1}{\sum_{i=1}^k \gamma_i}) = O(\frac{1}{k^{1-q}})$, which is also as the same as SGD and subgradient algorithms for $\gamma_k = O(\frac{1}{k^q})$.

# 4 Convergence analysis for nonconvex minimization

This section considers the convergence of MCGD when one or more of $f_i$ is nonconvex. In this case, we assume $f_i$, $i = 1, 2, \ldots, M$, are differentiable and $\nabla f_i$ is Lipschitz with $L^2$. We also set $X$ as the full space. We study the following scheme

$$x^{k+1} = x^k - \gamma_k \nabla f_{j_k}(x^k). \tag{17}$$

We prove non-ergodic convergence of the expected gradient norm of $f$ under diminishing non-summable stepsizes. The stepsize requirements in this section are slightly stronger than those in the convex case with an extra $\ln k$ factor. In this part, we use the following assumption.

**Assumption 3** *The gradients of $f_i$ are assumed to be bounded, i.e., there exists $D > 0$ such that*

$$\|\nabla f_i(x)\| \leq D, \quad i \in [M]. \tag{18}$$

We use this new assumption because $X$ is now the full space, and we have to directly bound the size of $\|\nabla f_i(x)\|$. In the nonconvex case, we cannot obtain objective value convergence, and we only bound the gradients. Now, we are prepared to present our convergence results of nonconvex MCGD.

**Theorem 2** *Let Assumptions 1 and 3 hold and $(x^k)_{k \geq 0}$ be generated by scheme (17). Also, assume $f_i$ is differentiable and $\nabla f_i$ is L-Lipschitz, and the stepsizes satisfy*

$$\sum_k \gamma_k = +\infty, \sum_k \ln^2 k \cdot \gamma_k^2 < +\infty. \tag{19}$$

*Then, we have*

$$\lim_k \mathbb{E}\|\nabla f(x^k)\| = 0. \tag{20}$$

*and*

$$\mathbb{E}\big(\min_{1 \leq i \leq k}\{\|\nabla f(x^i)\|^2\}\big) = O\Big(\frac{\psi(P)}{\sum_{i=1}^k \gamma_i}\Big), \tag{21}$$

*where $\psi(P)$ is given in Lemma 1. If we select the stepsize as $\gamma_k = O(\frac{1}{k^q})$, $\frac{1}{2} < q < 1$, then we get the rate $\mathbb{E}\big(\min_{1 \leq i \leq k}\{\|\nabla f(x^i)\|^2\}\big) = O(\frac{\psi(P)}{k^{1-q}})$.*

*Furthermore, let $(e^k)_{k \geq 0}$ be a sequence of noise and consider the inexact nonconvex MCGD iteration:*

$$x^{k+1} = x^k - \gamma_k \big(\nabla f_{j_k}(x^k) + e^k\big). \tag{22}$$

*If the noise sequence obeys*

$$\sum_{k=1}^{+\infty} \gamma_k \cdot \|e^k\| < +\infty, \tag{23}$$

*then the convergence results (20) and (21) still hold for inexact nonconvex MCGD. In addition, if we set $\gamma_k = O(\frac{1}{k^q})$ as $\frac{1}{2} < q < 1$ and the noise satisfy $\|e^k\| = O(\frac{1}{k^p})$ for $p + q > 1$, then (20) still holds and $\mathbb{E}\big(\min_{1 \leq i \leq k}\{\|\nabla f(x^i)\|^2\}\big) = O(\frac{\psi(P)}{k^{1-q}})$.*

This proof of Theorem 2 is different from previous one. In particular, we cannot expect some sort of convergence to $f(x^*)$, where $x^* \in \operatorname{argmin} f$ due to nonconvexity. To this end, we use the Lipschitz continuity of $\nabla f_i$ ($i \in [M]$) to derive the "descent". Here, the "$O$" contains a polynomial compisition of constants $D$ and $L$.

Compared with MCGD in the convex case, the stepsize requirements of nonconvex MCGD become a tad higher; in summable part, we need $\sum_k \ln^2 k \cdot \gamma_k^2 < +\infty$ rather than $\sum_k \ln k \cdot \gamma_k^2 < +\infty$. Nevertheless, we can still use $\gamma_k = O(\frac{1}{k^q})$ for $\frac{1}{2} < q < 1$.

# 5 Convergence analysis for continuous state space

When the state space $\Xi$ is a continuum, there are infinitely many possible states. In this case, we consider an infinite-state Markov chain that is time-homogeneous and reversible. Using the results in [8, Theorem 4.9], the mixing time of this kind of Markov chain still has geometric decrease like (11). Since Lemma 1 is based on a linear algebra analysis, it no longer applies to the continuous case. Nevertheless, previous results still hold with nearly unchanged proofs under the following assumption:

**Assumption 4** *For any $\xi \in \Xi$, $|F(x;\xi) - F(y;\xi)| \le L\|x-y\|$, $\sup_{x \in X, \xi \in \Xi}\{\|\hat{\nabla} F(x;\xi)\|\} \le D$, $\mathbb{E}_\xi \hat{\nabla} F(x;\xi) \in \partial \mathbb{E}_\xi F(x;\xi)$, and $\sup_{x,y \in X, \xi \in \Xi} |F(x;\xi) - F(y;\xi)| \le H$.*

We consider the general scheme

$$x^{k+1} = \mathbf{Proj}_X\big(x^k - \gamma_k(\hat{\nabla} F(x^k;\xi^k) + e^k)\big), \tag{24}$$

where $\xi^k$ are samples on a Markov chain trajectory. If $e^k \equiv \mathbf{0}$, the scheme then reduces to (3).

**Corollary 1** *Assume $F(\cdot;\xi)$ is convex for each $\xi \in \Xi$. Let the stepsizes satisfy (12) and $(x^k)_{k \ge 0}$ be generated by Algorithm (24), and $(e^k)_{k \ge 0}$ satisfy (16). Let $F^* := \min_{x \in X} \mathbb{E}_\xi(F(x;\xi))$. If Assumption 4 holds and the Markov chain is time-homogeneous, irreducible, aperiodic, and reversible, then we have*

$$\lim_k \mathbb{E}\big(\mathbb{E}_\xi(F(x^k;\xi)) - F^*\big) = 0, \quad \mathbb{E}(\mathbb{E}_\xi(F(\overline{x^k};\xi)) - F^*) = O\Big(\frac{\max\{1, \frac{1}{\ln(1/\lambda)}\}}{\sum_{i=1}^k \gamma_i}\Big),$$

*where $0 < \lambda < 1$ is the geometric rate of the mixing time of the Markov chain (which corresponds to $\lambda(P)$ in the finite-state case).*

Next, we present our result for a possibly nonconvex objective function $F(\cdot;\xi)$ under the following assumption.

**Assumption 5** *For any $\xi \in \Xi$, $F(x;\xi)$ is differentiable, and $\|\nabla F(x;\xi) - \nabla F(y;\xi)\| \le L\|x-y\|$. In addition, $\sup_{x \in X, \xi \in \Xi}\{\|\nabla F(x;\xi)\|\} < +\infty$, $X$ is the full space, and $\mathbb{E}_\xi \nabla F(x;\xi) = \nabla \mathbb{E}_\xi F(x;\xi)$.*

Since $F(x,\xi)$ is differentiable and $X$ is the full space, the iteration reduces to

$$x^{k+1} = x^k - \gamma_k(\nabla F(x^k;\xi^k) + e^k). \tag{25}$$

**Corollary 2** *Let the stepsizes satisfy (19), $(x^k)_{k \ge 0}$ be generated by Algorithm (25), the noises obey (23), and Assumption 5 hold. Assume the Markov chain is time-homogeneous, irreducible, and aperiodic and reversible. Then, we have*

$$\lim_k \mathbb{E}\|\nabla \mathbb{E}_\xi(F(x^k;\xi))\| = 0, \quad \mathbb{E}\big(\min_{1 \le i \le k}\{\|\nabla \mathbb{E}_\xi(F(x^i;\xi))\|^2\}\big) = O\Big(\frac{\max\{1, \frac{1}{\ln(1/\lambda)}\}}{\sum_{i=1}^k \gamma_i}\Big), \tag{26}$$

*where $0 < \lambda < 1$ is geometric rate for the mixing time of the Markov chain.*

# 6 Conclusion

In this paper, we have analyzed the stochastic gradient descent method where the samples are taken on a trajectory of Markov chain. One of our main contributions is non-ergodic convergence analysis for convex MCGD, which uses a novel line of analysis. The result is then extended to the inexact gradients. This analysis lets us establish convergence for non-reversible finite-state Markov chains and for nonconvex minimization problems. Our results are useful in the cases where it is impossible or expensive to directly take samples from a distribution, or the distribution is not even known, but sampling via a Markov chain is possible. Our results also apply to decentralized learning over a network, where we can employ a random walker to traverse the network and minimizer the objective that is defined over the samples that are held at the nodes in a distribute fashion.

## Footnotes

[2]This is for the convenience of the presentation in the proofs. If each $f_i$ has a $L_i$, it is possible to improve our results slights. But, we simply set $L := \max_i\{L_i\}$

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
