[Supplementary Material]

# Supplementary material for *On Markov Chain Gradient Descent*

## 6.1 Technical lemmas

We present technical lemmas used in this paper.

**Lemma 2** *Consider two nonnegative sequences $(\alpha_k)_{k \geq 0}$ and $(h_k)_{k \geq 0}$ that satisfy*

1. $\lim_k h_k = 0$ *and* $\sum_k h_k = +\infty$, *and*

2. $\sum_k \alpha_k h_k < +\infty$, *and*

3. $|\alpha_{k+1} - \alpha_k| \leq ch_k$ *for some $c > 0$ and $k = 0, 1, \ldots$.*

*Then, we have $\lim \alpha_k = 0$.*

We call the sequence $(\alpha_k)_{k \geq 0}$ satisfying parts 1 and 2 a *weakly summable sequence* since it is not necessarily summable but becomes so after multiplying a non-summable yet diminishing sequence $h_k$. Without part 3, it is generally impossible to claim that $\alpha_k$ converges to 0. This lemma generalizes [15, Lemma 12].

**Proof of Lemma 2**

From parts 1 and 2, we have $\liminf_k \alpha_k = 0$. Therefore, it suffices to show $\limsup_k \alpha_k = 0$.

Assume $\limsup_k \alpha_k > 0$. Let $v := \frac{1}{3} \limsup_k \alpha_k > 0$. Then, we have *infinite many* segments $\alpha_k, \alpha_{k+1}, \ldots, \alpha_{k'}$ such that $k < k'$ and

$$\alpha_k < v \leq \alpha_{k+1}, \ldots, \alpha_{k'-1} \leq 2v < \alpha_{k'}. \tag{27}$$

It is possible that $k' = k + 1$, then, the terms $\alpha_{k+1}, \ldots, \alpha_{k'}$ in (27) will vanish. But it does not affect the following proofs. By the assumption $|\alpha_{k+1} - \alpha_k| \leq ch_k \to 0$, we further have $\frac{v}{2} < \alpha_k < v \leq \alpha_{k+1}$ for infinitely many *sufficiently large* $k$. This leads to the following contradiction

$$\sum_{j=k}^{k'-1} h_j = h_k + \sum_{j=k+1}^{k'-1} h_j \leq \frac{2\alpha_k}{v} h_k + \sum_{j=k+1}^{k'-1} \frac{\alpha_j}{v} h_j \leq \frac{2}{v} \sum_{j=k}^{k'-1} \alpha_k h_k \to 0, \tag{28}$$

$$\sum_{j=k}^{k'-1} h_j \geq \frac{1}{c} \sum_{j=k}^{k'-1} |\alpha_{j+1} - \alpha_j| \geq \frac{1}{c} \sum_{j=k}^{k'-1} (\alpha_{j+1} - \alpha_j) = \frac{1}{c}(\alpha_{k'} - \alpha_k) > \frac{v}{c}. \tag{29}$$

The following lemma is used to derive the boundedness of some specific sequence. It is used in the inexact MCGD.

**Lemma 3** *Consider four nonnegative sequences $(\alpha_k)_{k \geq 0}$, $(\eta_k)_{k \geq 0} \in \ell^1$ and $(\epsilon_k)_{k \geq 0} \in \ell^1$ that satisfy*

$$\alpha_{k+1} + h_k \leq (1 + \eta_k)\alpha_k + \epsilon_k. \tag{30}$$

*Then, we have $(h_k)_{k \geq 0} \in \ell^1$ and $\sum_k h_k = O(\max\{\sum_k \epsilon_k, (e^{\sum_k \eta_k} \cdot \sum_k \eta_k), (e^{\sum_k \eta_k} \cdot \sum_k \eta_k \cdot \sum_k \epsilon_k)\})$.*

**Proof of Lemma 3**

The convergence Lemma 3 has been given in [12, Theorem 1]. Here, we prove the order for $\sum_k h_k$. Noting that $h_k \geq 0$, we then have

$$\alpha_{k+1} \leq (1 + \eta_k)\alpha_k + \epsilon_k.$$

As we have nonnegative number sequences $1 + \eta_k \le e^{\eta_k}$, so

$$
\begin{aligned}
\alpha_{k+1} &\le (1 + \eta_k)\alpha_k + \epsilon_k \\
&\le e^{\eta_k}\alpha_k + \epsilon_k \\
&\le e^{\eta_k + \eta_{k-1}}\alpha_{k-1} + e^{\eta_k}\epsilon_{k-1} + \epsilon_k \\
&\vdots \\
&\le e^{\sum_{i=1}^{k} \eta_i}\alpha_1 + e^{\sum_{i=1}^{k} \eta_i} \cdot \sum_{i=1}^{k} \epsilon_i.
\end{aligned}
\tag{31}
$$

Thus, we get $\alpha_k = O(\max\{e^{\sum_{i=1}^{k} \eta_i}, e^{\sum_{i=1}^{k} \eta_i} \cdot \sum_{i=1}^{k} \epsilon_i\})$. With direct calculations, we get

$$
\sum_k h_k \le \sum_k (\alpha_k - \alpha_{k+1}) + \sup_k\{\alpha_k\} \sum_k \eta_k + \sum_k \epsilon_k
$$

Using the got estimation of $\alpha_k$, we then derive the result.

**Lemma 4** *Let $a > b > 0$, and $c > 0$, and $n \ge 0$ be real numbers. Then,*

$$
cx^n b^x \le a^x
\tag{32}
$$

*if $x \ge \max\{0, \frac{(2n+2)(\ln c + n \ln(\frac{2n+2}{\ln a/b}) - n)}{(n+2)\ln(a/b)}\}$.*

**Proof of Lemma 4**

Let $\ell := \frac{a}{b}$, then, we just need to consider the function

$$
D(x) := x \cdot \ln\ell - n \cdot \ln x - \ln c.
\tag{33}
$$

Letting $x_0 = \frac{2n+2}{\ln\ell}$ and the convexity of $-\ln(x)$ when $x > 0$,

$$
-n \cdot \ln x \ge -\frac{n}{x_0}(x - x_0) - n \cdot \ln x_0 = -\frac{n \ln\ell}{2n+2}x + n - n \cdot \ln x_0.
\tag{34}
$$

Thus, we have

$$
D(x) \ge \frac{(n+2)\ln\ell}{2n+2}x + n - n \cdot \ln x_0 - \ln c \ge 0.
\tag{35}
$$

**Lemma 5** *Let $a > 0$, and $x > 0$ be a enough large real number. If*

$$
y - a\ln y + c = x.
\tag{36}
$$

*Then, it holds*

$$
y - x \le 2a\ln x.
\tag{37}
$$

**Proof of Lemma 5**

It is easy to see as $x$ is large, $y$ is very large. And then, (36) indicates the $y$ is actually an implicit function respect with $x$. Using the implicit function theorem,

$$
y'(x) = \frac{1}{1 - \frac{a}{y}}.
\tag{38}
$$

With L'Hospital's rule,

$$
\lim_{x \to +\infty} \frac{y - x}{\ln x} = \lim_{x \to +\infty} \frac{y'(x) - 1}{\frac{1}{x}} = \lim_{x \to +\infty} \frac{ax}{y - a} = \lim_{y \to +\infty} \frac{ay - a^2\ln y + ac}{y - a} = a.
\tag{39}
$$

Then, as $x$ is large enough,

$$
\frac{y - x}{\ln x} \le 2a.
\tag{40}
$$

**Proof of Lemma 1**

With direct calculation, for any $A, B \in \mathbb{C}^{M \times M}$, we have

$$\|AB\|_F \leq \|A\|_F \|B\|_F.$$

Since $P$ is a *convergent matrix*[3], it is known from [9] that the Jordan normal form of $P$ is

$$P = U \begin{bmatrix} 1 & & & \\ & J_2 & & \\ & & \ddots & \\ & & & J_d \end{bmatrix} U^{-1}, \tag{41}$$

where $d$ is the number of the blocks, $n_i \geq 1$ is the dimension of the $i$th block submatrix $J_i$, $i = 2, 3, \ldots, d$, which satisfy $\sum_{i=1}^{d} n_i = M$, and matrix $J_i := \lambda_i(P) \cdot \mathbb{I}_{n_i} + \mathbb{D}(-1, n_i)$ with

$$\mathbb{D}(-1, n_i) := \begin{bmatrix} 0 & 1 & & \\ & \ddots & \ddots & \\ & & \ddots & 1 \\ & & & 0 \end{bmatrix}_{n_i \times n_i} \quad \text{and } \mathbb{I}_{n_i} \text{ being the identity matrix of size } n_i. \text{ By Assumption}$$

1, we have $\lambda_1(P) = 1$ and $|\lambda_i(P)| < 1$, $i = 2, 3, \ldots, M$. Through direct calculations, we have

$$P^k = U \begin{bmatrix} 1 & & & \\ & J_2^k & & \\ & & \ddots & \\ & & & J_d^k \end{bmatrix} U^{-1}.$$

Let $C_k^l := \binom{k}{l}$ for $0 \leq l \leq k$, and $C_k^l := 0$ for $0 \leq k < l$. For $i = 2, 3, \ldots, d$, we directly calculate:

$$J_i^k = (\lambda_i(P) \cdot \mathbb{I}_{n_1} + \mathbb{D}(-1, n_i))^k = \sum_{l=0}^{k} C_k^l (\lambda_i(P))^{k-l} (\mathbb{D}(-1, n_i))^l$$

$$= \begin{bmatrix} (\lambda_i(P))^k & (\lambda_i(P))^{k-1} C_k^1 & (\lambda_i(P))^{k-2} C_k^2 & \cdots & (\lambda_i(P))^{k-n_i+1} C_k^{n_i-1} \\ & (\lambda_i(P))^k & (\lambda_i(P))^{k-1} C_k^1 & \ddots & \vdots \\ & & \ddots & \ddots & \vdots \\ & & & (\lambda_i(P))^k & (\lambda_i(P))^{k-1} C_k^1 \\ & & & & (\lambda_i(P))^k \end{bmatrix}_{n_i \times n_i}.$$

For $j = 0, 1, \ldots, n_i - 1$, we have

$$\left| (\lambda_i(P))^{k-j} C_k^j \right| \leq \left| (\lambda_i(P)) \right|^{k-n_i+1} C_k^j \leq |\lambda_2(P)|^{k-n_i+1} k^{n_i-1}.$$

With the technical Lemma 4 in Appendix, if $k \geq \max \left\{ \left\lceil \frac{2n_i(n_i-1)(\ln(\frac{2n_i}{\ln \lambda(P)/\lambda_2(P)})-1)}{(n_i+1)\ln(\lambda(P)/\lambda_2(P))} \right\rceil, 0 \right\}$, we further have

$$|\lambda_2(P)|^{k-n_i+1} k^{n_i-1} \leq \lambda^k(P). \tag{42}$$

Hence, for $i = 2, 3, \ldots, d$, we have $\lim_k J_i^k = \mathbf{0}$ and, thus,

$$\Pi^* = \lim_k P^k = U \begin{bmatrix} 1 & & & \\ & 0 & & \\ & & \ddots & \\ & & & 0 \end{bmatrix} U^{-1}.$$

For the sake of convenience, let $G^k := \begin{bmatrix} 0 & & & \\ & J_2^k & & \\ & & \ddots & \\ & & & J_d^k \end{bmatrix}$. Observing $\delta^k = \Pi^* - P^k = UG^kU^{-1}$,

$$\|\delta^k\|_\infty \le \|\delta^k\|_F = \|UG^kU^{-1}\|_F \le \|U\|_F\|U^{-1}\|_F \cdot \|G^k\|_F. \tag{43}$$

Based on the structure of $G^k$ and (42),

$$\|G^k\|_F \le \big(\sum_{i=2}^d n_i^2\big)^{\frac{1}{2}} \cdot \lambda^k(P). \tag{44}$$

Substituting (44) into (43), we the get

$$C_P := \big(\sum_{i=2}^d n_i^2\big)^{\frac{1}{2}} \cdot \|U\|_F\|U^{-1}\|_F \tag{45}$$

and

$$K_P := \max\Big\{\max_{1\le i\le d}\Big\{\Big\lceil\frac{2n_i(n_i-1)(\ln(\frac{2n_i}{\ln\lambda(P)/|\lambda_2(P)|})-1)}{(n_i+1)\ln(\lambda(P)/|\lambda_2(P)|)}\Big\rceil\Big\}, 0\Big\}. \tag{46}$$

## 6.2 Notation

The following notation is used through the proofs

$$\Delta^k := x^{k+1} - x^k. \tag{47}$$

For function $f$ and set $X$, $f^*$ denotes the minimum value of $f$ over $X$. In this paper, we assume that the stationary state of Markov chain is uniform, i.e., $\pi^* = (\frac{1}{M}, \cdots, \frac{1}{M})$.

**Proposition 1** *Let $(x^k)_{k\ge0}$ be generated by convex MCGD (5). For any $x^*$ being the minimizer of $f$ constrained on $X$, and $i \in [M]$, and $\forall k \in \mathbb{N}$, there exist some $H > 0$ such that*

1. $\|v\| \le D$, $\forall v \in \partial f_i(x^k)$, *and*

2. $|f_i(x^k) - f_i(x^*)| \le H$, *and*

3. $| f_i(x) - f_i(y) | \le D \cdot \|x - y\|$, $\forall x, y \in X$, *and*

4. $\|\Delta^k\| \le D \cdot \gamma_k$.

**Proof of Proposition 1**

The boundedness of $X$ gives a bound on $(x^k)_{k\ge0}$ based on the scheme of convex MCGD (5). With the convexity of $f_i$, $i = 1, 2, \ldots, M$, [Theorem 10.4, [13]] tells us

$$D := \sup_{v\in\partial f_i(x), x\in X, i\in[M]}\{\|v\|\} < +\infty.$$

Items 1, 2 and 4 are directly derived from the boundedness of the sequence and the set $X$. Item 3 is due to the convexity of $f_i$, which gives us

$$\langle v_1, x - y\rangle \le f_i(x) - f_i(y) \le \langle v_2, x - y\rangle,$$

where $v_1 \in \partial f(x)$ and $v_2 \in \partial f(y)$. With the Cauchy inequality, we are then led to

$$| f_i(x) - f_i(y) | \le \max\{\|v_1\|, \|v_2\|\} \cdot \|x - y\| \le D \cdot \|x - y\|. \tag{48}$$

*Though this following proofs, we use the following sigma algebra*

$$\chi^k := \sigma(x^1, x^2, \ldots, x^k, j_0, j_1, \ldots, j_{k-1}).$$

## 6.3 Proof of Theorem 1, the part for exact MCGD

We first prove (14) in Part 1 and then (13) in Part 2.

**Part 1. Proof of** (14). For any $x^*$ minimizing $f$ over $X$, we can get

$$\|x^{k+1} - x^*\|^2 \overset{a)}{=} \|\mathbf{Proj}_X(x^k - \gamma_k \hat{\nabla} f_{j_k}(x^k)) - \mathbf{Proj}_X(x^*)\|^2$$

$$\overset{b)}{\leq} \|x^k - \gamma_k \hat{\nabla} f_{j_k}(x^k) - x^*\|^2$$

$$\overset{c)}{=} \|x^k - x^*\|^2 - 2\gamma_k \langle x^k - x^*, \hat{\nabla} f_{j_k}(x^k) \rangle + \gamma_k^2 \|\hat{\nabla} f_{j_k}(x^k)\|^2$$

$$\overset{d)}{\leq} \|x^k - x^*\|^2 - 2\gamma_k(f_{j_k}(x^k) - f_{j_k}(x^*)) + \gamma_k^2 D^2, \tag{49}$$

where $a)$ uses the fact $x^* \in X$, $b)$ holds since $X$ is convex, $c)$ is direct expansion, and $d)$ follows from the convexity of $f_{j_k}$. Rearranging (49) and summing it over $k$ yield

$$\sum_k \gamma_k(f_{j_k}(x^k) - f_{j_k}(x^*)) \leq \frac{1}{2} \sum_k \left( \|x^k - x^*\|^2 - \|x^{k+1} - x^*\|^2 \right) + \frac{D^2}{2} \sum_k \gamma_k^2 \tag{50}$$

$$\leq \frac{1}{2}\|x^0 - x^*\|^2 + \frac{D^2}{2} \sum_k \gamma_k^2, \tag{51}$$

where the right side is non-negative and finite. For simplicity, let

$$\sum_k \gamma_k(f_{j_k}(x^k) - f_{j_k}(x^*)) =: C_1 < +\infty. \tag{52}$$

For integer $k \geq 1$, denote the integer $\mathcal{J}_k$ as

$$\mathcal{J}_k := \min\{\max\left\{ \left\lceil \ln\left( \frac{k}{2C_P H} \right) / \ln(1/\lambda(P)) \right\rceil, K_P \right\}, k\}. \tag{53}$$

$\mathcal{J}_k$ is important to the analysis and frequently used. Obviously, $\mathcal{J}_k \leq k$; this is because we need use $x^{k-\mathcal{J}_k}$ in the following. With Lemma 1 and direct calculations, we have

$$\mid [P^{\mathcal{J}_k}]_{i,j} - \frac{1}{M} \mid \leq C_P(\lambda(p))^{\mathcal{J}_k} \leq \frac{1/k}{2H}, \text{ for any } i,j \in \{1,2,\ldots,M\}. \tag{54}$$

The remaining of Part 1 consists of two steps:

1. in Step 1, we will prove $\sum_k \gamma_k \mathbb{E}(f(x^{k-\mathcal{J}_k}) - f^*) \leq C_4 + \frac{C_5}{\ln(1/\lambda(P))}, C_4, C_5 > 0$ and

2. in Step 2, we will show $\sum_k \gamma_k \mathbb{E}(f(x^k) - f(x^{k-\mathcal{J}_k})) \leq C_6 + \frac{C_7}{\ln(1/\lambda(P))}, C_6, C_7 > 0$.

Then, summing them gives us

$$\sum_k \gamma_k \mathbb{E}(f(x^k) - f^*) = O(\max\{1, \frac{1}{\ln(1/\lambda(P))}\}). \tag{55}$$

and, by convexity of $f$ and Jensen's inequality,

$$(\sum_{i=1}^{k} \gamma_i) \cdot \mathbb{E}(f(\overline{x^k}) - f^*) \leq \sum_{i=1}^{k} \gamma_i \mathbb{E}(f(x^i) - f^*) = O(\max\{1, \frac{1}{\ln(1/\lambda(P))}\}) < +\infty. \tag{56}$$

Rearrangement of (56) then gives us (14).

**Step 1:** We can get

$$\gamma_k \mathbb{E}[(f_{j_k}(x^{k-\mathcal{J}_k}) - f_{j_k}(x^k))] \overset{a)}{\leq} C\gamma_k \mathbb{E}\|x^{k-\mathcal{J}_k} - x^k\|$$

$$\overset{b)}{\leq} C\gamma_k \sum_{d=k-\mathcal{J}_k}^{k-1} \mathbb{E}\|\Delta^d\| \overset{c)}{\leq} CD \sum_{d=k-\mathcal{J}_k}^{k-1} \gamma_d \gamma_k$$

$$\overset{d)}{\leq} \frac{CD}{2} \sum_{d=k-\mathcal{J}_k}^{k-1} (\gamma_d^2 + \gamma_k^2) = \frac{CD}{2} \mathcal{J}_k \gamma_k^2 + \frac{CD}{2} \sum_{d=k-\mathcal{J}_k}^{k-1} \gamma_d^2, \tag{57}$$

where $a$) follows from (48), $b$) uses the triangle inequality, $c$) uses Proposition 1, Part 4, and $d$) applies the Schwartz inequality. From Lemma 1, we can see

$$\mathscr{J}_k = O(\frac{\ln k}{\ln(1/\lambda(P))}). \tag{58}$$

From the assumption on $\gamma_k$, it follows that $(\mathscr{J}_k \gamma_k^2)_{k \geq 0}$ is summable.

Next, we establish the summability of $\sum_{d=k-\mathscr{J}_k}^{k-1} \gamma_d^2$ over $k$. We consider an integer $K$ large enough that activates Lemma 5, and can let $\mathscr{J}_k = \frac{\ln(2C_P H \cdot k)}{\ln(1/\lambda(P))}$ when $k \geq K$. Noting that finite items do not affect the summability of sequence, we then turn to studying $(\sum_{d=k-\mathscr{J}_k}^{k-1} \gamma_d^2)_{k \geq K}$. For any $k' \geq K$, $\gamma_{k'}^2$ appears at most $\sharp\{t \in \mathbb{Z}^+ \mid t - \mathscr{J}_t \leq k' \leq t, K \leq t\}$ times in the summation $\sum_{k=K}^{+\infty} \sum_{d=k-\mathscr{J}_k}^{k-1} \gamma_d^2$. Let $t(k)$ be the solution of $t - \mathscr{J}_t = k$. The direct computation tells us

$$\sharp\{t \in \mathbb{Z}^+ \mid t - \mathscr{J}_t \leq k \leq t, K \leq t\} \leq t(k) - k \leq 2\frac{\ln k}{\ln(1/\lambda(P))},$$

where the last inequality is due to Lemma 5. Therefore,

$$\sum_{k=K}^{+\infty} \Big( \sum_{d=k-\mathscr{J}_k}^{k-1} \gamma_d^2 \Big) \leq \frac{2}{\ln(1/\lambda(P))} \sum_{k=K}^{+\infty} \ln k \cdot \gamma_k^2 < +\infty. \tag{59}$$

Since both terms in the right-hand side of (57) are finite, we conclude $\sum_{k=0}^{+\infty} \gamma_k \mathbb{E}[f_{j_k}(x^{k-\mathscr{J}_k}) - f_{j_k}(x^k)] < +\infty$. Combining (52), it then follows

$$\sum_k \gamma_k \mathbb{E}(f_{j_k}(x^{k-\mathscr{J}_k}) - f_{j_k}(x^*)) \leq C_2 + \frac{C_3}{\ln(1/\lambda(P))} \tag{60}$$

for some $C_2, C_3 > 0$. Due to that finite items have no effect on the summability. In the following, similarly, we assume $k \geq K_P$.

Recall $\chi^k := \sigma(x^1, x^2, \ldots, x^k, j_0, j_1, \ldots, j_{k-1})$. We derive an important lower bound

$$\begin{aligned}
\mathbb{E}_{j_k}\big(f_{j_k}(x^{k-\mathscr{J}_k}) - f_{j_k}(x^*)) \mid \chi^{k-\mathscr{J}_k}\big) &\stackrel{a)}{=} \sum_{i=1}^{M} \big(f_i(x^{k-\mathscr{J}_k}) - f_i(x^*)\big) \cdot \mathbb{P}(j_k = i \mid \chi^{k-\mathscr{J}_k}) \\
&\stackrel{b)}{=} \sum_{i=1}^{M} \big(f_i(x^{k-\mathscr{J}_k}) - f_i(x^*)\big) \cdot \mathbb{P}(j_k = i \mid j_{k-\mathscr{J}_k}) \\
&\stackrel{c)}{=} \sum_{i=1}^{M} (f_i(x^{k-\mathscr{J}_k}) - f_i(x^*)) \cdot [P^{\mathscr{J}_k}]_{j_{k-\mathscr{J}_k}, i} \\
&\stackrel{d)}{\geq} (f(x^{k-\mathscr{J}_k}) - f^*) - \frac{1}{2k},
\end{aligned} \tag{61}$$

where $a$) is the definition of the conditional expectation, and $b$) uses the Markov property, and $c$) follows from $\mathbb{P}(j_k = i \mid j_{k-\mathscr{J}_k}) = [P^{\mathscr{J}_k}]_{j_{k-\mathscr{J}_k}, i}$, and $d$) is due to (54). Taking total expectations of (61) and multiplying by $\gamma_k$, switching the sides then yields

$$\gamma_k \mathbb{E}(f(x^{k-\mathscr{J}_k}) - f^*) \leq \gamma_k \mathbb{E}(f_{j_k}(x^{k-\mathscr{J}_k}) - f_{j_k}(x^*)) + \frac{\gamma_k}{2k}. \tag{62}$$

Combining (60) and (62) and using

$$\sum_{k \geq 1} \frac{\gamma_k}{k} \leq \frac{1}{2} \sum_{k \geq 1} \gamma_k^2 + \frac{1}{2} \sum_{k \geq 1} \frac{1}{k^2} < +\infty,$$

we arrive at

$$\sum_k \gamma_k \mathbb{E}(f(x^{k-\mathscr{J}_k}) - f^*) \leq C_4 + \frac{C_5}{\ln(1/\lambda(P))} \tag{63}$$

for some $C_4, C_5 > 0$.

**Step 2:** With direct calculation (the same procedure as (57)), we get

$$\gamma_k \cdot \mathbb{E}(f(x^k) - f(x^{k-\mathcal{J}_k})) \leq \frac{MD^2}{2} \mathcal{J}_k \gamma_k^2 + \frac{MD^2}{2} \sum_{d=k-\mathcal{J}_k}^{k-1} \gamma_d^2, \tag{64}$$

where $M$ is number of the finite functions. The summability of $(\mathcal{J}_k \gamma_k^2)_{k\geq 1}$ and $(\sum_{d=k-\mathcal{J}_k}^{k-1} \gamma_d^2)_{k\geq 1}$ has been proved, thus,

$$\sum_k \gamma_k \cdot \mathbb{E}(f(x^k) - f(x^{k-\mathcal{J}_k})) \leq C_6 + \frac{C_7}{\ln(1/\lambda(P))}$$

for some $C_6, C_7 > 0$.

Now, we prove the Part 2.

**Part 2. Proof of** (13). Using the Lipschitz continuity of $f$ and Proposition 1, we have
$$|f(x^{k+1}) - f(x^k)| \leq D\|\Delta^k\| \leq D^2 \cdot \gamma_k \tag{65}$$

and, thus,

$$\begin{aligned} &|\mathbb{E}(f(x^{k+1}) - f^*) - \mathbb{E}(f(x^k) - f^*)| \\ &= |\mathbb{E}(f(x^{k+1}) - f(x^k))| \leq \mathbb{E}|f(x^{k+1}) - f(x^k)| \leq D^2 \cdot \gamma_k. \end{aligned} \tag{66}$$

From (55), (66), and Lemma 2 (letting $\alpha_k = \mathbb{E}(f(x^k) - f^*)$ and $h_k = \gamma_k$ in Lemma 2), we then get

$$\lim_k \mathbb{E}(f(x^k) - f^*) = 0. \tag{67}$$

## 6.4 Proof of Theorem 1, the part for inexact MCGD (15)

For any $x^*$ that minimizes $f$ over $X$, we have

$$\begin{aligned} \|x^{k+1} - x^*\|^2 &\overset{a)}{=} \|\mathbf{Proj}_X(x^k - \gamma_k \hat{\nabla} f_{j_k}(x^k) - \gamma_k e^k) - \mathbf{Proj}_X(x^*)\|^2 \\ &\overset{b)}{\leq} \|x^k - \gamma_k \hat{\nabla} f_{j_k}(x^k) - \gamma_k e^k - x^*\|^2 \\ &\overset{c)}{=} \|x^k - x^*\|^2 - 2\gamma_k \langle x^k - x^*, \hat{\nabla} f_{j_k}(x^k) \rangle \\ &\quad + 2\gamma_k \langle x^* - x^k, e^k \rangle + \gamma_k^2 \|\hat{\nabla} f_{j_k}(x^k) + e^k\|^2 \\ &\overset{d)}{\leq} (1 + \ln k \cdot \gamma_k^2)\|x^k - x^*\|^2 - 2\gamma_k (f_{j_k}(x^k) - f_{j_k}(x^*)) \\ &\quad + 2\gamma_k^2 D^2 + 2\gamma_k^2 \|e^k\|^2 + \frac{\|e^k\|^2}{\ln k}, \end{aligned} \tag{68}$$

where $a)$ uses the fact $x^* \in X$, $b)$ uses the convexity of $X$, $c)$ applies direct expansion, and $d)$ uses the Schwartz inequality $2\gamma_k \langle x^* - x^k, e^k \rangle \leq \ln k \cdot \gamma_k^2 \cdot \|x^* - x^k\|^2 + \frac{\|e^k\|^2}{\ln k}$. Taking expectations on both sides, we then get

$$\begin{aligned} \mathbb{E}\|x^{k+1} - x^*\|^2 &+ 2\mathbb{E}(\gamma_k(f_{j_k}(x^k) - f_{j_k}(x^*))) \\ &\leq (1 + \ln k \cdot \gamma_k^2)\mathbb{E}\|x^k - x^*\|^2 + 2\gamma_k^2 D^2 + 2\gamma_k^2 \|e^k\|^2 + \frac{\|e^k\|^2}{\ln k}. \end{aligned} \tag{69}$$

Following the same deductions in the proof in §6.3, we can get

$$\gamma_k \mathbb{E}(f(x^k) - f^*) \leq \mathbb{E}(\gamma_k(f_{j_k}(x^k) - f_{j_k}(x^*))) + w^k, \tag{70}$$

where $(w^k)_{k\geq 0} \in \ell^1$ is a nonnegative, summable sequence that is defined as certain weighted sums out of $(\gamma_k)_{k\geq 0}$ and it is easy to how $\sum_k w^k = O(\max\{1, \frac{1}{\ln(1/\lambda(P))}\})$. Then, we can obtain

$$\begin{aligned} \mathbb{E}\|x^{k+1} - x^*\|^2 &+ 2\gamma_k \mathbb{E}(f(x^k) - f^*) \\ &\leq (1 + \ln k \cdot \gamma_k^2)\mathbb{E}\|x^k - x^*\|^2 + 2\gamma_k^2 D^2 + 2\gamma_k^2 \|e^k\|^2 + \frac{\|e^k\|^2}{\ln k} + w^k. \end{aligned} \tag{71}$$

After applying Lemma 3 to (71), we get

$$\sum_k \gamma_k \mathbb{E}(f(x^k) - f^*) = O(\max\{1, \frac{1}{\ln(1/\lambda(P))}\}) < +\infty. \tag{72}$$

The remaining of this proof is very similar to the proof in §6.3.

### 6.5 Proof of Theorem 2, the part for exact nonconvex MCGD (17)

With Assumption 3, we can get the following fact.

**Proposition 2** *Let $(x^k)_{k \geq 0}$ be generated by nonconvex MCGD (17). It then holds that*

$$\|\Delta^k\| \leq D \cdot \gamma_k. \tag{73}$$

We first prove (21) in Part 1 and then (20) in Part 2.

**Part 1. Proof of** (21). For integer $k \geq 1$, denote the integer $\mathcal{T}_k$ as

$$\mathcal{T}_k := \min\{\max\left\{ \left\lceil \ln\left(\frac{k}{2C_P D^2}\right)/\ln(\frac{1}{\lambda(P)})\right\rceil, K_P \right\}, k\}. \tag{74}$$

By using Lemma 1, we then get

$$\left| [P^{\mathcal{T}_k}]_{i,j} - \frac{1}{M} \right| \leq \frac{1/k}{2D^2}, \text{ for any } i, j \in \{1, 2, \ldots, M\}. \tag{75}$$

The remaining of Part 1 consists of two major steps:

1. in first step, we will prove $\sum_k \gamma_k \mathbb{E} \|\nabla f(x^{k-\mathcal{T}_k})\|^2 = O(\max\{1, \frac{1}{\ln(1/\lambda(P))}\})$, and

2. in second step, we will show $\sum_k \left( \gamma_k \mathbb{E} \|\nabla f(x^k)\|^2 - \gamma_k \mathbb{E} \|\nabla f(x^{k-\mathcal{T}_k})\|^2 \right) \leq C_3 + \frac{C_4}{\ln(1/\lambda(P))}, C_3, C_4 > 0$.

Summing them together, we are led to

$$\sum_k \gamma_k \mathbb{E} \|\nabla f(x^k)\|^2 = O(\max\{1, \frac{1}{\ln(1/\lambda(P))}\}). \tag{76}$$

With direct calculations, we then get

$$(\sum_{i=1}^{k} \gamma_i) \cdot \mathbb{E}(\min_{1 \leq i \leq k} \{\|\nabla f(x^i)\|^2\}) \leq \sum_{i=1}^{k} \gamma_i \mathbb{E} \|\nabla f(x^i)\|^2 = O(\max\{1, \frac{1}{\ln(1/\lambda(P))}\}) < +\infty. \tag{77}$$

Rearrangement of (77) then gives us (21).

**Step 1.** The direct computations give the following lower bound:

$$\mathbb{E}_{j_k}(\langle \nabla f(x^{k-\mathcal{T}_k}), \nabla f_{j_k}(x^{k-\mathcal{T}_k}) \rangle \mid \chi^{k-\mathcal{T}_k})$$

$$\stackrel{a)}{=} \sum_{i=1}^{M} \langle \nabla f(x^{k-\mathcal{T}_k}), \nabla f_{j_k}(x^{k-\mathcal{T}_k}) \rangle \cdot \mathbb{P}(j_k = i \mid \chi^{k-\mathcal{T}_k})$$

$$\stackrel{b)}{=} \sum_{i=1}^{M} \langle \nabla f(x^{k-\mathcal{T}_k}), \nabla f_{j_k}(x^{k-\mathcal{T}_k}) \rangle \cdot \mathbb{P}(j_k = i \mid j_{k-\mathcal{T}_k})$$

$$\stackrel{c)}{=} \sum_{i=1}^{M} \langle \nabla f(x^{k-\mathcal{T}_k}), \nabla f_i(x^{k-\mathcal{T}_k}) \rangle \cdot [P^{\mathcal{T}_k}]_{j_{k-\mathcal{T}_k}, i}$$

$$\stackrel{d)}{\geq} \|\nabla f(x^{k-\mathcal{T}_k})\|^2 - \frac{1}{2k}, \tag{78}$$

where $a)$ is from the conditional expectation, and $b)$ depends on the property of Markov chain, and $c)$ is the matrix form of the probability, and $d)$ is due to (75). Rearrangement of (78) gives us

$$\gamma_k \mathbb{E}\|\nabla f(x^{k-\mathcal{T}_k})\|^2 \leq \gamma_k \mathbb{E}(\langle \nabla f(x^{k-\mathcal{T}_k}), \nabla f_{j_k}(x^{k-\mathcal{T}_k})\rangle) + \frac{\gamma_k}{2k}. \tag{79}$$

We present the bound of $f(x^{k+1}) - f(x^k)$ as

$$f(x^{k+1}) - f(x^k) \overset{a)}{\leq} \langle \nabla f(x^k), \Delta^k \rangle + \frac{L\|\Delta^k\|^2}{2}$$

$$\overset{b)}{=} \langle \nabla f(x^{k-\mathcal{T}_k}), \Delta^k \rangle + \langle \nabla f(x^k) - \nabla f(x^{k-\mathcal{T}_k}), \Delta^k \rangle + \frac{L\|\Delta^k\|^2}{2}$$

$$\overset{c)}{\leq} \langle \nabla f(x^{k-\mathcal{T}_k}), \Delta^k \rangle + \frac{(L+1)\|\Delta^k\|^2}{2} + \frac{L^2\|x^k - x^{k-\mathcal{T}_k}\|^2}{2}, \tag{80}$$

where $a)$ uses continuity of $\nabla f$, and $b)$ is a basic algebra computation, $c)$ applies the Schwarz inequality to $\langle \nabla f(x^k) - \nabla f(x^{k-\mathcal{T}_k}), \Delta^k \rangle$. Moving $\langle \nabla f(x^{k-\mathcal{T}_k}), \Delta^k \rangle$ to left side, we then get

$$\langle \nabla f(x^{k-\mathcal{T}_k}), -\Delta^k \rangle \leq f(x^k) - f(x^{k+1}) + \frac{L^2\|x^{k+1} - x^{k-\mathcal{T}_k}\|^2}{2} + \frac{(L+1)\|\Delta^k\|^2}{2}. \tag{81}$$

We turn to offering the following bound:

$$\mathbb{E}(\langle \nabla f(x^{k-\mathcal{T}_k}), -\Delta^k \rangle \mid \chi^{k-\mathcal{T}_k})$$

$$= \gamma_k \mathbb{E}(\langle \nabla f(x^{k-\mathcal{T}_k}), \nabla f_{j_k}(x^k)\rangle \mid \chi^{k-\mathcal{T}_k})$$

$$= \gamma_k \mathbb{E}(\langle \nabla f(x^{k-\mathcal{T}_k}), \nabla f_{j_k}(x^{k-\mathcal{T}_k})\rangle \mid \chi^{k-\mathcal{T}_k})$$

$$+ \gamma_k \mathbb{E}(\langle \nabla f(x^{k-\mathcal{T}_k}), \nabla f_{j_k}(x^k) - \nabla f_{j_k}(x^{k-\mathcal{T}_k})\rangle \mid \chi^{k-\mathcal{T}_k})$$

$$\geq \gamma_k \mathbb{E}(\langle \nabla f(x^{k-\mathcal{T}_k}), \nabla f_{j_k}(x^{k-\mathcal{T}_k})\rangle \mid \chi^{k-\mathcal{T}_k})$$

$$- D \cdot L \cdot \mathbb{E}(\gamma_k \|x^k - x^{k-\mathcal{T}_k}\| \mid \chi^{k-\mathcal{T}_k}), \tag{82}$$

where we used the Lipschitz continuity and boundedness of $\nabla f$. Taking conditional expectations on both sides of (81) on $\chi^{k-\mathcal{T}_k}$ and rearrangement of (82) tell us

$$\gamma_k \mathbb{E}_{j_k}(\langle \nabla f(x^{k-\mathcal{T}_k}), \nabla f_{j_k}(x^{k-\mathcal{T}_k})\rangle \mid \chi^{k-\mathcal{T}_k})$$

$$\leq \mathbb{E}(f(x^k) - f(x^{k+1}) \mid \chi^{k-\mathcal{T}_k}) + \frac{(L+1) \cdot \mathbb{E}(\|\Delta^k\|^2 \mid \chi^{k-\mathcal{T}_k})}{2}$$

$$+ D \cdot L \cdot \mathbb{E}(\gamma_k \|x^k - x^{k-\mathcal{T}_k}\| \mid \chi^{k-\mathcal{T}_k}) + \frac{L^2 \cdot \mathbb{E}(\|x^{k+1} - x^{k-\mathcal{T}_k}\|^2 \mid \chi^{k-\mathcal{T}_k})}{2}. \tag{83}$$

Taking expectations on both sides of (83), we then get

$$\gamma_k \mathbb{E}(\langle \nabla f(x^{k-\mathcal{T}_k}), \nabla f_{j_k}(x^{k-\mathcal{T}_k})\rangle) \leq \underbrace{\mathbb{E}(f(x^k) - f(x^{k+1}))}_{\text{(I)}} + \underbrace{\frac{(L+1) \cdot \mathbb{E}\|\Delta^k\|^2}{2}}_{\text{(II)}}$$

$$+ \underbrace{D \cdot L \cdot \mathbb{E}(\gamma_k \|x^k - x^{k-\mathcal{T}_k}\|)}_{\text{(III)}} + \underbrace{\frac{L^2 \cdot \mathbb{E}(\|x^{k+1} - x^{k-\mathcal{T}_k}\|^2)}{2}}_{\text{(IV)}}. \tag{84}$$

We now prove that (I), (II), (III) and (IV) are all summable. The summability (I) is obvious. For (II), (III) and (IV), with Proposition 2, we can derive (we omit the constant parameters in following)

$$\text{(II)} : \mathbb{E}(\|\Delta^k\|^2) \leq \gamma_k^2 D^2,$$

and

$$\text{(III)} : \mathbb{E}(\gamma_k \|x^k - x^{k-\mathcal{T}_k}\|) \leq \gamma_k \sum_{d=k-\mathcal{T}_k}^{k-1} \mathbb{E}\|\Delta^d\| \leq D \sum_{d=k-\mathcal{T}_k}^{k-1} \gamma_d \gamma_k$$

$$\leq \frac{D}{2} \sum_{d=k-\mathcal{T}_k}^{k-1} (\gamma_d^2 + \gamma_k^2) = \frac{\mathcal{T}_k D}{2} \gamma_k^2 + \frac{D}{2} \sum_{d=k-\mathcal{T}_k}^{k-1} \gamma_d^2,$$

and

$$(\text{IV}): \mathbb{E}(\|x^{k+1} - x^{k-\mathcal{T}_k}\|^2) \le (\mathcal{T}_k + 1) \sum_{d=k-\mathcal{T}_k}^{k} \mathbb{E}\|\Delta^d\|^2 \le D^2(\mathcal{T}_k + 1) \sum_{d=k-\mathcal{T}_k}^{k} \gamma_d^2.$$

It is easy to see if $(\mathcal{T}_k \sum_{d=k-\mathcal{T}_k}^{k} \gamma_d^2)_{k \ge 0}$ is summable, (II), (III) and (IV) are all summable. We consider a large enough integer $K$ which makes Lemma 5 active, and $\mathcal{T}_k = \frac{\ln(2C_P D^2 \cdot k)}{\ln(1/\lambda(P))}$ when $k \ge K$. Noting that finite items do not affect the summability of sequence, we then turn to studying $(\mathcal{T}_k \sum_{d=k-\mathcal{J}_k}^{k-1} \gamma_d^2)_{k \ge K}$. For any fixed integer $t \ge K$, $\gamma_t^2$ only appears at index $k \ge K$ satisfying

$$S_t := \{k \in \mathbb{Z}^+ \mid k - \mathcal{T}_k \le t \le k-1, \ k \ge K\}$$

in the inner summation. If $K$ is large enough, $\mathcal{T}_k \le \frac{k}{2}$, and then

$$k \le 2t, \ \ \forall k \in S_t.$$

Noting that $\mathcal{T}_k$ increases respect to $k$, we then get

$$\mathcal{T}_k \le \mathcal{T}_{2t}, \ \ \forall k \in S_t.$$

That means in $\sum_{k=K}^{+\infty}(\mathcal{T}_k \sum_{d=k-\mathcal{T}_k}^{k-1} \gamma_d^2)$, $\gamma_t^2$ appears at most

$$\mathcal{T}_{2t} \cdot \sharp(S_t) \le \frac{2\ln^2 t}{\ln(1/\lambda^2(P))} + \frac{2\ln t}{\ln(1/\lambda^2(P))} + \frac{2\ln(2C_P D^2)}{\ln(1/\lambda^2(P))}.$$

The direct computation then yields

$$\sum_{k=K}^{+\infty} \left( \mathcal{T}_k \sum_{d=k-\mathcal{T}_k}^{k-1} \gamma_d^2 \right) \le \frac{2}{\ln(1/\lambda^2(P))} \sum_{t=K} \ln^2 t \cdot \gamma_t^2$$

$$+ \frac{2}{\ln(1/\lambda^2(P))} \sum_{t=K} \ln t \cdot \gamma_t^2 + \frac{2\ln(2C_P D^2)}{\ln(1/\lambda^2(P))} \sum_{t=K} \gamma_t^2 = O(\frac{1}{\ln(1/\lambda(P))}). \tag{85}$$

Turning back to (84), we then get

$$\sum_k \gamma_k \mathbb{E}(\langle \nabla f(x^{k-\mathcal{T}_k}), \nabla f_{j_k}(x^{k-\mathcal{T}_k}) \rangle) \le C_1 + \frac{C_2}{\ln(1/\lambda(P))},$$

for some $C_1, C_2 > 0$. By using (79),

$$\sum_k \gamma_k \mathbb{E}\|\nabla f(x^{k-\mathcal{T}_k})\|^2 = O(\max\{1, \frac{1}{\ln(1/\lambda(P))}\}). \tag{86}$$

**Step 2:** With Lipschitz of $\nabla f$, we can do the following basic algebra

$$\gamma_k \|\nabla f(x^k)\|^2 - \gamma_k \|\nabla f(x^{k-\mathcal{T}_k})\|^2$$
$$\le \gamma_k \langle \nabla f(x^k) - \nabla f(x^{k-\mathcal{T}_k}), \nabla f(x^k) + \nabla f(x^{k-\mathcal{T}_k}) \rangle$$
$$\le \gamma_k \|\nabla f(x^k) - \nabla f(x^{k-\mathcal{T}_k})\| \cdot \|\nabla f(x^k) + \nabla f(x^{k-\mathcal{T}_k})\|$$
$$\le 2DL\gamma_k \|x^k - x^{k-\mathcal{T}_k}\| \le DL\gamma_k^2 + DL\|x^k - x^{k-\mathcal{T}_k}\|^2. \tag{87}$$

We have proved $(\mathbb{E}\|x^{k+1} - x^{k-\mathcal{T}_k}\|^2)_{k \ge 0}$ is summable $(O(\max\{1, \frac{1}{\ln(1/\lambda(P))}\}))$; it is same way to prove that $(\mathbb{E}\|x^k - x^{k-\mathcal{T}_k}\|^2)_{k \ge 0}$ is summable $(O(\max\{1, \frac{1}{\ln(1/\lambda(P))}\}))$. Thus,

$$\sum_k \left( \gamma_k \mathbb{E}\|\nabla f(x^k)\|^2 - \gamma_k \mathbb{E}\|\nabla f(x^{k-\mathcal{T}_k})\|^2 \right) \le C_3 + \frac{C_4}{\ln(1/\lambda(P))}$$

for some $C_3, C_4 > 0$.

**Part 2. Proof of** (20). With the Lipschitz continuity of $\nabla f$, we have

$$\big| \ \|\nabla f(x^{k+1})\|^2 - \|\nabla f(x^k)\|^2 \ \big| \le 2DL\|\Delta^k\| \le 2DL \cdot \gamma_k. \tag{88}$$

That is also

$$\left| \mathbb{E} \|\nabla f(x^{k+1})\|^2 - \mathbb{E} \|\nabla f(x^k)\|^2 \right| \leq 2DL \cdot \gamma_k \tag{89}$$

With (76), (89), and Lemma 2 (letting $\mathbb{E}\|\nabla f(x^{k+1})\|^2 = \alpha_k$ and $\gamma_k = h_k$ in Lemma 2), it follows

$$\lim_k \mathbb{E}\|\nabla f(x^k)\|^2 = 0. \tag{90}$$

With Schwarz inequality

$$(\mathbb{E}\|\nabla f(x^k)\|)^2 \leq \mathbb{E}\|\nabla f(x^k)\|^2,$$

the result is then proved.

## 6.6   Proof of Theorem 2, the part for inexact nonconvex MCGD (22)

The proof is very similar to §6.5 except several places. We first modify (82) as

$$\begin{aligned}
\mathbb{E}(\langle \nabla f(x^{k-\mathcal{T}_k}), -\Delta^k \rangle \mid \chi^{k-\mathcal{T}_k}) \\
&= \gamma_k \mathbb{E}(\langle \nabla f(x^{k-\mathcal{T}_k}), \nabla f_{j_k}(x^k)\rangle \mid \chi^{k-\mathcal{T}_k}) + \gamma_k \mathbb{E}(\langle \nabla f(x^{k-\mathcal{T}_k}), e^k \rangle \mid \chi^{k-\mathcal{T}_k}) \\
&= \gamma_k \mathbb{E}(\langle \nabla f(x^{k-\mathcal{T}_k}), \nabla f_{j_k}(x^{k-\mathcal{T}_k})\rangle \mid \chi^{k-\mathcal{T}_k}) \\
&\quad + \gamma_k \mathbb{E}(\langle \nabla f(x^{k-\mathcal{T}_k}), \nabla f_{j_k}(x^k) - \nabla_{j_k}(x^{k-\mathcal{T}_k})\rangle \mid \chi^{k-\mathcal{T}_k}) \\
&\quad + \gamma_k \mathbb{E}(\langle \nabla f(x^{k-\mathcal{T}_k}), e^k \rangle \mid \chi^{k-\mathcal{T}_k}) \\
&\geq \gamma_k \mathbb{E}(\langle \nabla f(x^{k-\mathcal{T}_k}), \nabla f_{j_k}(x^{k-\mathcal{T}_k})\rangle \mid \chi^{k-\mathcal{T}_k}) \\
&\quad - D \cdot L \cdot \mathbb{E}(\gamma_k \|x^k - x^{k-\mathcal{T}_k}\| \mid \chi^{k-\mathcal{T}_k}) - D \cdot \gamma_k \cdot \|e^k\|, \tag{91}
\end{aligned}$$

where we used the Lipschitz continuity and boundedness of $\nabla f$. And then taking expectations, we are then led to

$$\gamma_k \mathbb{E}(\langle \nabla f(x^{k-\mathcal{T}_k}), \nabla f_{j_k}(x^{k-\mathcal{T}_k})\rangle) \leq (\mathrm{I}) + (\mathrm{II}) + (\mathrm{III}) + (\mathrm{IV}) + D \cdot \gamma_k \cdot \|e^k\|, \tag{92}$$

where (I), (II), (III) and (IV) are given by (84). The following is the same as §6.5.

**Proof of Corollaries 1 and 2**

The proofs of Corollaries 1 and 2 are similar to previous. To give the credit to the reader, we just prove Corollary 1 in the exact case, i.e. $e^k \equiv \mathbf{0}$.

Let $F^* := \min_{x \in X} \mathbb{E}_\xi F(x; \xi)$. Like previous methods, the proof consists of two parts: in the first one, we prove $\sum_k \gamma_k \cdot \mathbb{E}(\mathbb{E}_\xi F(x^k; \xi) - F^*) < +\infty$; while in the second one, we focus on proving $|\mathbb{E}(\mathbb{E}_\xi(x^{k+1}; \xi) - F^*) - \mathbb{E}_\xi(F(x^k; \xi) - F^*)| = O(\gamma_k)$.

**Part 1.** For any $x^*$ minimizing $f$ over $X$, we can get

$$\begin{aligned}
\|x^{k+1} - x^*\|^2 &\overset{a)}{=} \|\mathbf{Proj}_X(x^k - \gamma_k \hat{\nabla} F(x^k; \xi^k)) - \mathbf{Proj}_X(x^*)\|^2 \\
&\overset{b)}{\leq} \|x^k - \gamma_k \hat{\nabla} f_{j_k}(x^k) - x^*\|^2 \\
&\overset{c)}{=} \|x^k - x^*\|^2 - 2\gamma_k \langle x^k - x^*, \hat{\nabla} F(x^k; \xi^k)\rangle + \gamma_k^2 \|\hat{\nabla} F(x^k; \xi^k)\|^2 \\
&\overset{d)}{\leq} \|x^k - x^*\|^2 - 2\gamma_k (F(x^k; \xi^k) - F(x^*; \xi^k)) + \gamma_k^2 D^2, \tag{93}
\end{aligned}$$

where $a)$ uses the fact $x^* \in X$, and $b)$ depends the concentration of operator $\mathbf{Proj}_X(\cdot)$ when $X$ is convex, and $c)$ is direct expansion, and $d)$ comes from the convexity of $F(x; \xi^k)$. Rearrangement of (93) tells us

$$\sum_k \gamma_k (F(x^k; \xi^k) - F(x^*; \xi^k)) \leq \frac{1}{2} \sum_k \left( \|x^k - x^*\|^2 - \|x^{k+1} - x^*\|^2 \right) + \frac{D^2}{2} \sum_k \gamma_k^2. \tag{94}$$

Noting the right side of (94) is non-negative and finite, we then get

$$\sum_k \gamma_k (F(x^k; \xi^k) - F(x^*; \xi^k)) = C_1, \tag{95}$$

for some $0 < C_1 < +\infty$. For integer $k \geq 1$, denote the integer $\mathcal{H}_k$ as

$$\mathcal{H}_k := \min\Big\{\Big\lceil \ln\Big(\frac{k}{2CH}\Big)\Big/\ln(1/\lambda)\Big\rceil, k\Big\}. \tag{96}$$

Here $C$ and $\lambda$ are constants which are dependent on the Markov chain. These notation are to give the difference to $C_P$ and $\lambda(P)$ in Lemma 1. Obviously, $\mathcal{H}_k \leq k$. With [8, Theorem 4.9] and direct calculations, we have

$$\int_\Xi |p_s^{s+\mathcal{H}_k}(\xi) - \pi(\xi)|d\mu(\xi) \leq \frac{1}{2 \cdot H \cdot k}, \forall s \in \mathbb{Z}^+ \tag{97}$$

where $p_s^{s+\mathcal{H}_k}(\xi)$ denotes the transition p.d.f. from $s$ to $s + \mathcal{H}_k$ with respect to $\xi$. Noting the Markov chain is time-homogeneous, $p_s^{s+\mathcal{H}_k}(\xi) = p_0^{\mathcal{H}_k}(\xi)$.

The remaining of Part 1 consists of two major steps:

1. in first step, we will prove $\sum_k \gamma_k \mathbb{E}(\mathbb{E}_\xi F(x^{k-\mathcal{H}_k}; \xi) - F^*) \leq C_4 + \frac{C_5}{\ln(1/\lambda)}$, $C_4, C_5 > 0$ and

2. in second step, we will show $\sum_k \gamma_k \mathbb{E}(\mathbb{E}_\xi F(x^k; \xi) - \mathbb{E}_\xi F(x^{k-\mathcal{H}_k}; \xi)) \leq C_6 + \frac{C_7}{\ln(1/\lambda)}$, $C_6, C_7 > 0$.

Summing them together, we are then led to

$$\sum_k \gamma_k \mathbb{E}(\mathbb{E}_\xi F(x^k; \xi) - F^*) = O(\max\{1, \frac{1}{\ln(1/\lambda)}\}). \tag{98}$$

With direct calculations, we are then led to

$$\Big(\sum_{i=1}^k \gamma_i\Big) \cdot \mathbb{E}(\mathbb{E}_\xi F(\overline{x^k}; \xi) - F^*) \leq \sum_{i=1}^k \gamma_i \mathbb{E}(\mathbb{E}_\xi F(x^i; \xi) - F^*) = O(\max\{1, \frac{1}{\ln(1/\lambda)}\}) < +\infty. \tag{99}$$

Rearrangement of (56) then gives us (14).

In the following, we prove these two steps.

**Step 1:** We can get

$$\gamma_k \mathbb{E}[F(x^{k-\mathcal{H}_k}; \xi^k) - F(x^k; \xi^k)] \overset{a)}{\leq} L\gamma_k \mathbb{E}\|x^{k-\mathcal{H}_k} - x^k\|$$

$$\overset{b)}{\leq} L\gamma_k \sum_{d=k-\mathcal{H}_k}^{k-1} \mathbb{E}\|\Delta^d\| \overset{c)}{\leq} DL \sum_{d=k-\mathcal{H}_k}^{k-1} \gamma_d \gamma_k$$

$$\overset{d)}{\leq} \frac{DL}{2} \sum_{d=k-\mathcal{H}_k}^{k-1} (\gamma_d^2 + \gamma_k^2) = \frac{DL}{2}\mathcal{H}_k \gamma_k^2 + \frac{DL}{2}\sum_{d=k-\mathcal{H}_k}^{k-1} \gamma_d^2, \tag{100}$$

where $a)$ comes from (48), $b)$ is the triangle inequality, $c)$ depends on Assumption 5, and $d)$ is from the Schiwarz inequality. As $k$ is large, we can see

$$\mathcal{H}_k = O(\frac{\ln k}{\ln(1/\lambda)}). \tag{101}$$

Recall the following proved inequality,

$$\sum_{k=K}^{+\infty}\Big(\sum_{d=k-\mathcal{H}_k}^{k-1} \gamma_d^2\Big) \leq \frac{2}{\ln(1/\lambda)}\sum_{k=K}^{+\infty} \ln k \cdot \gamma_k^2. \tag{102}$$

Turning back to (100), we can see $\sum_{k=0}^{+\infty} \gamma_k \mathbb{E}[F(x^{k-\mathcal{H}_k}; \xi^k) - F(x^k; \xi^k)] < +\infty$. Combining (95), it then follows

$$\sum_k \gamma_k \mathbb{E}(F(x^{k-\mathcal{H}_k}; \xi^k) - F(x^*; \xi^k)) \leq C_2 + \frac{C_3}{\ln(1/\lambda)} \tag{103}$$

for some $C_2, C_3 > 0$.

We consider the lower bound

$$\mathbb{E}_{\xi^k}(F(x^{k-\mathcal{H}_k}; \xi^k) - F(x^*; \xi^k)) \mid \chi^{k-\mathcal{H}_k})$$

$$\overset{a)}{=} \int_\Xi (F(x^{k-\mathcal{H}_k}; \xi) - F(x^*; \xi)) p_{k-\mathcal{H}_k}^k(\xi) d\mu(\xi)$$

$$\overset{b)}{=} \int_\Xi (F(x^{k-\mathcal{H}_k}; \xi) - F(x^*; \xi)) p_0^{\mathcal{H}_k}(\xi) d\mu(\xi)$$

$$\overset{c)}{\geq} (\mathbb{E}_\xi F(x^{k-\mathcal{H}_k}; \xi) - F^*) - \frac{1}{2k} \tag{104}$$

where $a)$ is from the conditional expectation, and $b)$ depends on the property of Markov chain, and $c)$ is due to (97). Taking expectations of (61) and multiplying by $\gamma_k$, switching the sides then yields

$$\gamma_k \mathbb{E}(\mathbb{E}_\xi F(x^{k-\mathcal{H}_k}; \xi) - F^*) \leq \gamma_k \mathbb{E}(\mathbb{E}_\xi F(x^{k-\mathcal{H}_k}; \xi^k) - \mathbb{E}_\xi F(x^*; \xi^k)) + \frac{\gamma_k}{2k}. \tag{105}$$

Substituting (103) into(105) and noting

$$\sum_{k \geq 1} \frac{\gamma_k}{k} \leq \frac{1}{2} \sum_{k \geq 1} \gamma_k^2 + \frac{1}{2} \sum_{k \geq 1} \frac{1}{k^2} < +\infty,$$

we are then led to

$$\sum_k \gamma_k \mathbb{E}(\mathbb{E}_\xi F(x^{k-\mathcal{H}_k}; \xi) - F^*) \leq C_4 + \frac{C_5}{\ln(1/\lambda)} \tag{106}$$

for some $C_4, C_5 > 0$.

**Step 2:** With direct calculation (the same procedure as (100)), we get

$$\gamma_k \cdot \mathbb{E}(\mathbb{E}_\xi F(x^k; \xi) - \mathbb{E}_\xi F(x^{k-\mathcal{H}_k}; \xi)) \leq \frac{D^2}{2} \mathcal{H}_k \gamma_k^2 + \frac{D^2}{2} \sum_{d=k-\mathcal{H}_k}^{k-1} \gamma_d^2. \tag{107}$$

The summability of $(\mathcal{H}_k \gamma_k^2)_{k \geq 1}$ and $(\sum_{d=k-\mathcal{H}_k}^{k-1} \gamma_d^2)_{k \geq 1}$ has been proved, thus,

$$\sum_k \gamma_k \cdot \mathbb{E}(\mathbb{E}_\xi F(x^k; \xi) - \mathbb{E}_\xi F(x^{k-\mathcal{H}_k}; \xi)) \leq C_6 + \frac{C_7}{\ln(1/\lambda(P))}$$

for some $C_6, C_7 > 0$.

Now, we prove the Part 2.

**Part 2.** With the Lipschitz continuity of $F(x, \xi)$, it follows

$$|\mathbb{E}_\xi F(x^{k+1}; \xi) - \mathbb{E}_\xi F(x^k; \xi)| \leq L\|\Delta^k\| \leq DL \cdot \gamma_k. \tag{108}$$

That is also

$$|\mathbb{E}(\mathbb{E}_\xi F(x^{k+1}; \xi) - F^*) - \mathbb{E}(\mathbb{E}_\xi F(x^k; \xi) - F^*)|$$

$$= |\mathbb{E}(\mathbb{E}_\xi F(x^{k+1}; \xi) - \mathbb{E}_\xi F(x^k; \xi))|$$

$$\leq \mathbb{E}|\mathbb{E}_\xi F(x^{k+1}; \xi) - \mathbb{E}_\xi F(x^k; \xi)| \leq DL \cdot \gamma_k. \tag{109}$$

With (98), (109), and Lemma 2 (letting $\alpha_k = \mathbb{E}(\mathbb{E}_\xi F(x^{k+1}; \xi) - F^*)$ and $h_k = \gamma_k$ in Lemma 2), we then get

$$\lim_k \mathbb{E}(\mathbb{E}_\xi F(x^k; \xi) - F^*)) = 0. \tag{110}$$

## Footnotes

[3]A matrix is *convergent* if its infinite power is convergent.