[Reviews · NeurIPS 2018]

Reviewer 1



There are great many analyses of stochastic gradient descent. The present paper studies sampling w.r.t. to a Markov chain and parametrising the rate of convergence with a proxy for the mixing time, which has been studied previously [3]. Clearly, one could replace the whole SGD e.g., with the famous hit-and-run Markov chain (at least for convex bodies), and some balance between the complexity of sampling the MC and the iteration complexity of the SGD using it would be of interest. The constants in the "O(n^4)" mixing-time result of Lovasz [1] are exorbitant, though. (It would seem like a shame to consider trajectories of lengths in the millions so as to decide on how to sample the gradient.) The contribution of the authors is to -- study the use of all samples in the trajectory, rather than only the first sample beyond the mixing time (upper bound). -- consider non-reversible (but still time-homogeneous, irreducible, and aperiodic) MC. Although I like the the general idea, there are three issues in the proof of the main theorem. In Eq. 53, line 382, there is a typo. Instead of ln(k / 2 C_P H), there should be ln(2 C_P H k). In Eq. 61, line 299, there is a typo. Instead of 1/2k, there should be M/2k. Much more importantly, the inequality in Eq. 54, line 384, holds only for sufficiently large k. In particular, (54) holds only if k >= K_P, which is the assumption in Lemma 1. Since the authors write "With Lemma 1 and...", this may be a language issue and they actual mean that they make the same assumption. But in (61), (62), (63), the authors do not assume k >= K_P any longer, while they use (54). Consequently, step d) in Eq. 61, line 399, does not work and (unless they sink the whole lot prior to mixing time into C_4) the whole proof fails. The authors should explain in the rebuttal, whether the proof is wrong, the C_4 is exorbitant, or what is the story. Minor comments: 1. There is no clear motivating application "carried through". Initially, the authors mention the counting of discrete solutions. One could imagine many other applications, but the actual experimental evaluation is performed on a trivial example. 2. Also, The definitions are not paritcularly clear or standard. The distance between probability distributions considered in the mixing time, where TV, Lp, and and various divergencies and entropies are used (c.f. 2.9-2.12 in [2]), is l\infty, which is not a common choice. 3. Further, the definition is obfuscated with some eigengap condition base don Fill's [4], which for MC of real interest is very likely not computable. Perhaps a parametrisation by the mixing time would do just as well? 4. Computationally, the paper is not convincing. The figures presented look ok, but when one looks at the number of parameters "in the first test problem", it is clear that there well may be many variants, for which the MC would not work as well. 5. Generally, the paper is poorly written. Just look at the introduction, there are numerous typos: "We aim to solving" -> "we aim to solve" "direct sampling from \Pi is expansive" -> "is expensive" "cardinal of" -> "cardinality of" "This is a huge save to the number of samples." -> "This presents huge savings in the number" [1] https://link.springer.com/article/10.1007/s101070050099 [2] https://people.cs.uchicago.edu/~laci/students/ganapathy.pdf [3] https://www.researchgate.net/profile/Mikael_Johansson6/publication/220133540_A_Randomized_Incremental_Subgradient_Method_for_Distributed_Optimization_in_Networked_Systems/links/0046352543f47cc8de000000.pdf [4] https://projecteuclid.org/euclid.aoap/1177005981 POST REBUTTAL: I do think that the edit to the proof suggested by the authors could work, but would lead to some exorbitant constant C4, a subject not addressed by the authors. Still, I have increased my score from "clear reject" to "accept" in the light of the fact that I am now happy with the validity of the proofs.

Reviewer 2



I'm happy with the authors' feedback, and will hence keep my "accept" decision. ============================================================== [Summary] This paper studies Markov chain gradient descent (MCGD) algorithm(s), an extension of stochastic gradient descent (SGD) in which the random samples are taken following a Markov chain. Built upon existing analysis of MCGD in [1,5,6], the paper generalizes the convergence result in various ways, answering a few open questions. In particular, the paper extends the ergodic result to non-ergodic ones, removes necessity for reversibility of the underlying Markov chain, and is able to deal with non-convex settings. The paper also provides a novel analysis based on keeping track of varying levels of mixing time corresponding to progressively better mixing, which may also shed light on future work. The numerical part of this paper is only for illustrative purposes (as the algorithm is not new), but the comparison between MCGD and SGD-T is informative and provides a good start and motivation for a more thorough study of the theoretical aspects of the algorithm. [Quality, originality and significance] The paper deals with a few challenging open problems in the analysis of MCGD. Previously, all analysis are done for convex objectives and reversible Markov chains. But non-reversible Markov chains can have much better mixing/convergence behavior both in theory and practice [14], and non-convex settings are of great practical importance in e.g. deep learning. The paper elegantly deals with these issues by introducing a new varying mixing time level analysis, without adding artificial assumptions. [Clarity] The paper is generally very well-written. The authors make notable efforts in motivating the necessity and challenges of the analysis for non-reversible and non-convex settings. In particular, the explanation of the assumptions between lines 117-128 is rather clear and frank. The example between lines 22-28 is also very good and showcase the advantage of MCGD over SGD in such a scenario. Nevertheless, a few things (mostly typos and wording) can still be slightly improved. Firstly, it might be slightly better to also mention the general MCMC in the example between lines 22-28. It may also help the readers to understand if a more explicit explanation of how the examples between lines 29-33 is related to the stochastic programming setting (1) is given. There are a few slight typos and wording issues, the removing of which may improve readability: 1. line 7: wider of -> wider range of; 2. line 19: expansive -> expensive; 3. line 25: If -> Even if; 4. line 29: natural -> naturally; 5. line 91: SG4 -> SGD4; 6. line 123: slower -> lower; 7. line 125: non-differential -> non-differentiable; 8. line 230: for the -> due to. Another suggestion is that the authors can explicitly state which results correspond to ergodic ones and which correspond to non-ergodic ones by e.g. directly stating that “Then we have the non-ergodic convergence” instead of just “Then we have” in line 193. This may help some readers who are not familiar with the definition of ergodicity in the optimization scenario to understand the results [Some suggestions for improvement] Here I list a few things that I think can be further improved: 1. The example between lines 61-68 does not seem to guarantee a uniform stationary distribution for the Markov chain over the graph, and hence may not be completely consistent with (4). The authors may want to pay some attention to this potential issue. 2. It may help a lot if the authors can provide a non-reversible Markov chain for the numerical experiments (or just prove and state, if the example on page 3 is), and provide non-ergodic convergence curves. This will make the motivating example on page 3 even more inspiring, which will then illustrate the good performance of MCGD beyond ergodic convergence, reversible Markov chains, and convexity, which naturally ask for the new analysis proposed later in this paper. 3. The authors may want to either mention at the end of Section 1 or the beginning of Section 2 that the optimization problem in study for the finite state cases is always in the form of (4). 4. Give at least one sentence of explanation for the significance of obtaining non-ergodic convergence results, either in theory or in practice. Also explain the major hurdle of restricting to reversible Markov chains and convex objectives in the previous works. In particular, the authors may want to highlight the main idea of how varying mixing levels help solve the non-convexity and non-reversibility issue, and how it leads to non-ergodic results. 5. It looks a bit weird to add the reversibility assumption back in Section 5 when continuous state space is involved. The authors may want to make more explanations for this. In general, this paper is well-written with insights and sufficient motivation, and it addresses a few challenging and open problems.

Reviewer 3



The paper demonstrates the convergence analysis of (stochastic) gradient descent methods under Markovian sample inputs. However, the experiment section is rather weak; it is from [1] but didn't compare with the method in [1]. Further, although the paper claim that their theoretical result for non-reversible Markovian chain would be substantially better at sampling, the experiment is not provided. Thus, I would recommend a rejection. Major comments: 1. (line 85) The loss may be undefined when the input to log < 0. Minor comments: 1. (Appendix eq 54) I do not understand why P^{J_k}_{i,j} converges to 1/M. Shouldn't it converge to \Pi_{ij}? 2. (Appendix ineq 61d) I do not see how eq (54) works here. Where is M and H? [1] Ergodic Mirror Descent. John Duchi et al. SIAM 2012. ==== POST REBUTTAL ==== I accept the explanation in the author feedback and bump the result to weak accept.